# INVGNN: Learning Invertible Node Representations on Graphs

**Giannis Nikolentzos** [* 1]   **Dimitrios Kelesis** [* 2]   **Nikolaos Nakis** [3]

## Abstract

Over the past decade, Graph Neural Networks (GNNs) have become a standard tool for solving machine learning problems on graphs. While many aspects of GNNs have been studied in depth, including their efficiency and expressive power, the invertibility of these models has remained largely unexplored. Standard aggregation functions, such as the mean, max and sum operators, are not invertible, which limits their applicability in tasks requiring invertible transformations. In this work, we introduce an invertible GNN layer. By stacking multiple such layers, we construct fully invertible GNN models, which we refer to as INVGNNs. These models inherit the benefits of invertible neural networks, including low memory usage for deep architectures, exact likelihood computation, and generative modeling capabilities. We demonstrate that INVGNNs can match the expressive power of the 1-dimensional Weisfeiler-Leman algorithm, showing that invertibility does not compromise model expressiveness. On standard graph classification benchmarks, our model performs comparably to other well-established GNNs. Beyond classification, we demonstrate the potential of invertible layers through density estimation tasks, including outlier detection and node feature generation.

## 1. Introduction

Graph-structured data is ubiquitous in several application domains including social networks, chemo-informatics and transportation science. Learning on graphs, however, presents significant challenges since traditional machine learning algorithms are typically designed for tabular or grid-like data, and are thus unable to model the complex dependencies between graph nodes.

The first neural networks for graphs were proposed several years ago (Frasconi et al., 1998; Gori et al., 2005; Scarselli et al., 2008). However, prior to the advent of deep learning, learning on graphs was mainly approached using graph kernels, which enabled the application of kernel-based methods, such as the SVM classifier, to graphs (Kriege et al., 2020; Nikolentzos et al., 2021). Over the past decade, neural networks have re-established themselves as the primary approach for performing machine learning on graph-structured data (Corso et al., 2024). These models, so-called Graph Neural Networks (GNNs), typically employ neighborhood aggregation mechanisms to learn node representations that incorporate both graph topology and node features. There is a wide variety of such models, each introducing different strategies for aggregating neighborhood information (Xu et al., 2019; Brody et al., 2022; Zhang et al., 2018). Adding a pooling layer after the neighborhood aggregation layers enables these architectures to produce representations for entire graphs.

GNNs have been studied intensively in recent years. Research has mainly focused on improving aggregation functions (Murphy et al., 2019; Vignac et al., 2020) and extending neighborhood definitions (Abu-El-Haija et al., 2019; Michel et al., 2023), designing more expressive models (Xu et al., 2019; Morris et al., 2019; Nikolentzos et al., 2020; Papp et al., 2021), enhancing scalability (Wu et al., 2019; Zeng et al., 2020), understanding models' generalization behavior (Morris et al., 2023; Brilliantov et al., 2024), explaining model decisions (Ying et al., 2019; Agarwal et al., 2022), and developing pooling layers for graph-level representations (Ying et al., 2018; Khasahmadi et al., 2020). In addition, substantial effort has been devoted to addressing fundamental challenges such as over-smoothing (Oono & Suzuki, 2020; Zhao & Akoglu, 2020) and over-squashing (Alon & Yahav, 2021; Topping et al., 2022). Despite the substantial body of work on GNNs, the invertibility of GNN layers has received little attention in prior work. Standard aggregation functions such as the mean and max operators fail to be injective (Xu et al., 2019). Consequently, neither of them is bijective. On the other hand, if the elements of the input multisets come from a countable set, the sum operator

---

[1]Department of Informatics and Telecommunications, University of Peloponnese, Tripoli, Greece [2]Institute of Informatics and Telecommunications, National Centre for Scientific Research "Demokritos", Athens, Greece [3]Human Nature Lab, Yale University, New Haven, USA. Correspondence to: Giannis Nikolentzos <nikolentzos@uop.gr>.

*Proceedings of the 43rd International Conference on Machine Learning*, Seoul, South Korea. PMLR 306, 2026. Copyright 2026 by the author(s).

can map the input multisets injectively to some Euclidean space. However, the sum of countably many vectors can only produce a subset of the Euclidean space and therefore, the operator is not bijective. Moreover, before aggregation, the multisets' elements need to be transformed by some function $f$ which is generally not bijective.

Invertible layers have attracted significant attention in the context of standard neural network architectures (Behrmann et al., 2019; Song et al., 2019; Behrmann et al., 2021). Unlike conventional layers, such as fully-connected or convolutional layers, which are generally not invertible and may discard information, invertible layers provide a one-to-one mapping between inputs and outputs. Networks built from invertible layers often exhibit low memory requirements, since activations of invertible layers do not need to be stored for backpropagation and can instead be recomputed on-the-fly (Gomez et al., 2017). These layers are particularly well-suited for applications such as density estimation, where the unknown density of input data can be mapped to a simple base distribution and the density can be computed using the change-of-variables formula (Dinh et al., 2017). Importantly, invertible layers lie at the core of normalizing flow architectures, a major family of generative models (Kobyzev et al., 2020; Papamakarios et al., 2021; Liu et al., 2019), which rely on sequences of invertible transformations to map simple probability distributions to complex data distributions. Sampling from these models can be performed by drawing from the simple distribution and applying the inverse transformation to obtain samples from the target distribution.

In this paper, we present an invertible layer for graph-structured data. By stacking a series of such layers, we can build invertible GNN models, referred to as INVGNNs. These models inherit all the advantages of invertible neural networks, such as low memory requirements for training very deep models, exact likelihood computation, and generative capabilities. To construct invertible layers, we use a neighborhood aggregation scheme built on an invertible graph operator and invertible fully-connected layers and activation functions. A major challenge in designing GNN layers is achieving sufficient expressive power. We show that the proposed model can be as expressive as the 1-dimensional Weisfeiler-Leman (1-WL) algorithm, demonstrating that invertibility does not come at the expense of expressive power. We evaluate the proposed model on standard graph classification tasks, where it performs on par with standard models of similar expressive power. The capabilities of the invertible layers are further illustrated through density estimation tasks for outlier detection, node feature generation, and interpretability. Our empirical results validate the model's ability to handle tasks where invertible layers are required. The code is available at https://github.com/giannisnik/invgnn.

## 2. Preliminaries

### 2.1. Notation

Let $\mathbb{N}$ denote the set of natural numbers, i.e., $\{1, 2, \ldots\}$. Then, $[n] = \{1, \ldots, n\} \subset \mathbb{N}$ for $n \geq 1$. Let also $\{\!\!\{\,\}\!\!\}$ denote a multiset, i.e., a generalized concept of a set that allows multiple instances for its elements. Let $G = (V, E)$ be an undirected graph, where $V$ is set of nodes and $E$ is the edge set. We will denote by $n$ the number of nodes and by $m$ the number of edges, i.e., $n = |V|$ and $m = |E|$. Let $\mathcal{N}(v)$ denote the neighborhood of node $v$, i.e., the set $\{u \mid (u, v) \in E\}$. The degree of a node $v$ is $\deg(v) = |\mathcal{N}(v)|$. The adjacency matrix $\mathbf{A} \in \mathbb{R}^{n \times n}$ encodes the edge information in a graph. The element of the $i$-th row and $j$-th column is equal to 1 if there is an edge between $v_i$ and $v_j$, and 0 otherwise. We use $\mathbf{H}^{(0)} \in \mathbb{R}^{n \times d}$ to denote the node features where $d$ is the dimension of the features. The feature $\mathbf{h}_{v_i}^{(0)}$ of node $v_i$ is stored in the $i$-th row of $\mathbf{H}^{(0)}$. We use $\mathbf{0}$, $\mathbf{1}$ and $\mathbf{I}$ to denote the vector of zeros, the vector of ones and the identity matrix, respectively. We denote by $\mathbf{X}_{i,j}$ the $(i, j)$-th entry of matrix $\mathbf{X}$. We also denote by $\mathbf{X}_{i,:}$ and $\mathbf{X}_{:,j}$ the $i$-th row and $j$-th column of $\mathbf{X}$, respectively.

### 2.2. Graph Neural Networks

As already discussed, standard GNNs employ a neighborhood aggregation scheme, where each node representation is updated based on the aggregation of its neighbors' representations. Let $\mathbf{h}_v^{(0)}$ denote node $v$'s initial feature vector. Then, for a number $K$ of iterations, standard GNNs update node representations as follows:

$$\mathbf{m}_v^{(k)} = \text{AGGREGATE}^{(k)}\left(\left\{\!\!\left\{\mathbf{h}_u^{(k-1)} \mid u \in \mathcal{N}(v)\right\}\!\!\right\}\right)$$

$$\mathbf{h}_v^{(k)} = \text{COMBINE}^{(k)}\left(\mathbf{h}_v^{(k-1)}, \mathbf{m}_v^{(k)}\right)$$

where AGGREGATE$^{(k)}$ is a permutation invariant function. By defining different AGGREGATE$^{(k)}$ and COMBINE$^{(k)}$ functions, we obtain different GNN instances. The employed neighborhood aggregation scheme is usually directly related to the expressive power of the model. Models that use the mean (Zhang et al., 2018) or weighted sum (with weights summing to 1) (Brody et al., 2022) functions as the AGGREGATE function are typically less powerful than 1-WL in distinguishing structurally different nodes (Nikolentzos et al., 2024). On the other hand, models that use the sum operator (Xu et al., 2019) as the AGGREGATE function can potentially match the expressiveness of 1-WL.

For node-level tasks, the final node representations $\mathbf{h}_v^{(K)}$ can be directly passed to a fully-connected layer (or a multi-layer perceptron (MLP)) for prediction. For graph-level tasks, a graph representation is obtained by aggregating the final representations of its nodes: $\mathbf{h}_G = \text{READOUT}\left(\{\!\!\{\mathbf{h}_v^{(K)} \mid v \in\right.$

$G\}\}$). The READOUT function is typically a differentiable permutation invariant function (e.g., sum, mean).

## 2.3. Change-of-variables Formula

The change-of-variables formula is the main mathematical mechanism underlying normalizing flows (Papamakarios et al., 2021), enabling the exact evaluation of probability densities for complex data distributions. Given a bijective, differentiable function $f \colon \mathcal{X} \to \mathcal{Z}$ that maps a complex data space $x$ to a well-defined latent space $z$, the formula relates the unknown data distribution $p_X(x)$ to a simple, tractable base distribution $p_Z(z)$ (such as a standard multivariate Gaussian) via the following equation:

$$p_X(x) = p_Z\big(f(x)\big) \left| \det \left( \frac{\partial f(x)}{\partial x} \right) \right| \qquad (1)$$

In this setting, the absolute value of the Jacobian determinant serves to measure the localized expansion or contraction of the probability volume caused by the transformation. Normalizing flows leverage this principle by composing multiple invertible neural network layers designed specifically to have highly expressive mappings but easily computable Jacobian determinants. Because this formulation preserves total probability without discarding information, there is no need for variational approximations or lower bounds, allowing for both exact maximum likelihood training and efficient sampling through the inverse mapping $f^{-1}(z)$.

## 3. The INVGNN Model

**Graph operators.** Neighborhood aggregation in GNN layers can be expressed as a simultaneous update of all node representations by multiplying the matrix of node features with a graph operator $\mathbf{S}$. In other words, the AGGREGATE and COMBINE functions are typically applied simultaneously to all graph nodes via some operator $\mathbf{S}$. For instance, the DGCNN model employs a mean aggregation operator (Zhang et al., 2018), given by: $\mathbf{S}_{\text{DGCNN}} = \tilde{\mathbf{D}}^{-1} \tilde{\mathbf{A}}$ where $\tilde{\mathbf{A}} = \mathbf{A} + \mathbf{I}$ and $\tilde{\mathbf{D}}$ is a diagonal matrix such that $\tilde{\mathbf{D}}_{ii} = \sum_{j=1}^{n} \tilde{\mathbf{A}}_{ij}$. The GIN model uses a graph operator that computes the sum of the representations of the neighbors and adds the node's own representation scaled by a learnable parameter (Xu et al., 2019): $\mathbf{S}_{\text{GIN}}^{(k)} = \mathbf{A} + \mathbf{I}\big(1 + \epsilon^{(k)}\big)$ where $\epsilon^{(k)}$ is a trainable scalar at layer $k$. The GCN model uses an operator that computes a weighted sum of the representations of the node itself and its neighbors, where the weights depend on the degrees of both the node and its neighbors (Kipf & Welling, 2017): $\mathbf{S}_{\text{GCN}} = \tilde{\mathbf{D}}^{-\frac{1}{2}} \tilde{\mathbf{A}} \tilde{\mathbf{D}}^{-\frac{1}{2}}$. Unfortunately, these operators are generally non-invertible for arbitrary graphs. Therefore, once the new node representations are computed, the previous representations cannot be uniquely recovered.

**Invertible graph operator.** We next introduce a graph operator for constructing invertible GNN layers. The operator is defined as the matrix exponential of the adjacency matrix:

$$\exp(\mathbf{A}) = \sum_{k=0}^{\infty} \frac{\mathbf{A}^k}{k!}$$

The element of the $i$-th row and $j$-th column of $\exp(\mathbf{A})$ is equal to the weighted sum of all walks of all lengths between nodes $v_i$ and $v_j$, with longer walks down-weighted by $1/k!$. Since walks can reach any node within the same connected component as the starting node, the matrix $\exp(\mathbf{A})$ consists of dense blocks, one for each connected component of the graph.

For undirected graphs, $\exp(\mathbf{A})$ can be computed in polynomial time as follows. We first compute the eigenvalue decomposition of $\mathbf{A}$. Since $\mathbf{A}$ is symmetric, we have that $\mathbf{A} = \mathbf{U}\mathbf{\Lambda}\mathbf{U}^{\top}$. Then, $\exp(\mathbf{A})$ is equal to:

$$\exp(\mathbf{A}) = \mathbf{U} \exp(\mathbf{\Lambda})\mathbf{U}^{\top}$$

where $\exp(\mathbf{\Lambda}) = \text{diag}(e^{\lambda_1}, \ldots, e^{\lambda_n})$ and $\lambda_1, \ldots, \lambda_n$ denote the eigenvalues of $\mathbf{A}$ (which are real for undirected graphs). Importantly, matrix $\exp(\mathbf{A})$ is known to be invertible and the inverse is equal to:

$$\big(\exp(\mathbf{A})\big)^{-1} = \exp(-\mathbf{A}) = \mathbf{U} \exp(-\mathbf{\Lambda})\mathbf{U}^{\top}$$

**Feature update.** To update the node representations, we apply a linear transformation to the current representations. Specifically, we use an invertible weight matrix to transform node representations. Such a layer is referred to as an invertible $1 \times 1$ convolution layer in computer vision (Kingma & Dhariwal, 2018). The weight matrix $\mathbf{W} \in \mathbb{R}^{d \times d}$ is initialized as a random rotation matrix, and is then parameterized using its LU decomposition:

$$\mathbf{W} = \mathbf{P}\,\mathbf{L}\big(\mathbf{U} + \text{diag}(\mathbf{s})\big)$$

where $\mathbf{P}$ is a permutation matrix, $\mathbf{L}$ is a lower triangular matrix with ones on the diagonal, $\mathbf{U}$ is an upper triangular matrix with zeros on the diagonal, and $\mathbf{s}$ is a vector. The permutation matrix $\mathbf{P}$ remains fixed during training, while matrices $\mathbf{L}, \mathbf{U}$ and vector $\mathbf{s}$ are optimized. The inverse transformation can be computed as follows:

$$\mathbf{W}^{-1} = \big(\mathbf{U} + \text{diag}(\mathbf{s})\big)^{-1}\mathbf{L}^{-1}\mathbf{P}^{\top}$$

**The INVGNN model.** Given the invertible graph operator and the invertible $1 \times 1$ convolution layer presented above, we define the node update rule of the proposed INVGNN model. For a number $K$ of iterations, the model updates node representations as follows:

$$\mathbf{h}_i^{(k)} = \sum_{j=1}^{n} \exp\big(\mathbf{A}\big)_{i,j} f\Big(\mathbf{h}_j^{(k-1)} \mathbf{W}^{(k)} + \mathbf{b}^{(k)}\Big) \qquad (2)$$

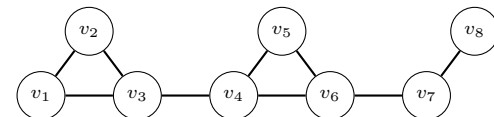

*Figure 1.* Counterexample for the proof of Proposition 3.1.

where $\exp\left(\mathbf{A}\right)_{i,j}$ denotes the $(i,j)$-th entry of the matrix $\exp(\mathbf{A})$ and $f$ is an invertible activation function (e.g., Tanh, LeakyReLU). In matrix notation, node representations are updated as follows:

$$\mathbf{H}^{(k)} = \exp\left(\mathbf{A}\right) f\left(\mathbf{H}^{(k-1)}\mathbf{W}^{(k)} + \mathbf{b}^{(k)}\right)$$

It is easy to show that the inverse mapping from layer $k+1$ back to layer $k$ can be computed as follows:

$$\mathbf{H}^{(k)} = \left(f^{-1}\left(\exp\left(\mathbf{A}\right)^{-1}\mathbf{H}^{(k+1)}\right) - \mathbf{b}^{(k+1)}\right)\left(\mathbf{W}^{(k+1)}\right)^{-1}$$

**Expressive power.** A model that consists of the above neighborhood aggregation layer is invertible, but we would also like it to be expressive enough (ideally at least as expressive as 1-WL). Since matrix $\exp(\mathbf{A})$ encodes relationships between nodes that are far from each other in the graph, it could be the case that a single layer of the proposed model possesses the same power as 1-WL. Unfortunately, it turns out that a single layer is less expressive than 1-WL (and therefore less expressive than GIN).

**Proposition 3.1.** *Let* $\mathbf{h} = \exp(\mathbf{A})\mathbf{1}$. *Let* $\mathbf{h}_i \in \mathbb{R}$ *denote the representation of node* $v_i$. *There exist nodes* $v_i, v_j$ *which are assigned different colors at some iteration* $k \in \mathbb{N}$ *of 1-WL, but for which* $\mathbf{h}_i = \mathbf{h}_j$ *holds.*

*Proof.* We use the graph illustrated in Figure 1, originally introduced by Powers & Sulaiman (1982), to prove the Proposition. For nodes $v_3$ and $v_6$, we have that $\mathbf{h}_3 = \mathbf{h}_6$. However, after three iterations, the 1-WL algorithm assigns different colors to nodes $v_3$ and $v_6$. □

On the contrary, the following Theorem shows that a multilayer INVGNN can be as powerful as 1-WL.

**Theorem 3.2.** *Let* $\mathcal{G}_{\leq n}(\Sigma)$ *be the collection of all unweighted, undirected graphs with at most* $n \in \mathbb{N}$ *vertices, where nodes are annotated with initial features from* $\Sigma \subset \mathbb{R}^d$, *where* $\Sigma$ *is a countable alphabet. Let* $c^{(k)}(v)$ *denote the color that 1-WL assigns to node* $v$ *after* $k$ *iterations. Consider an* INVGNN *model that consists of* $K$ *layers, and within each layer node representations are updated as in Equation* (2) *where* $f \colon \mathbb{R} \to \mathbb{R}$ *is an analytic non-polynomial function. Set* $d' = 2n + 1$. *Let* $v$ *and* $v'$ *be nodes in graphs* $G$ *and* $G'$, *respectively, where* $G, G' \in \mathcal{G}_{\leq n}(\Sigma)$. *Then for Lebesgue almost any* $\theta = \left(\mathbf{W}^{(1)} \in \mathbb{R}^{d \times d'}, \ldots, \mathbf{W}^{(K)} \in \mathbb{R}^{d' \times d'}, \mathbf{b}^{(1)} \in \mathbb{R}^{d'}, \ldots, \mathbf{b}^{(K)} \in \mathbb{R}^{d'}\right)$, $\mathbf{h}_v^{(K)} \neq \mathbf{h}_{v'}^{(K)}$ *if* $c^{(K)}(v) \neq c^{(K)}(v')$.

*Proof.* The proof relies on a recent result by Amir et al. (2023) and is provided in Appendix A. □

**Computational complexity.** The price to pay for the invertibility of the proposed layer is its increased time and memory complexity. A single layer of a standard GNN model has time complexity $\mathcal{O}(m)$, where $m$ is the number of edges. In contrast, a layer of the proposed model has time complexity $\mathcal{O}(n^2)$, since the matrix exponential $\exp(\mathbf{A})$ is dense for connected graphs. The memory complexity likewise scales as $\mathcal{O}(n^2)$. For dense graphs, where $m \approx n^2$, the time and space complexities of the proposed model are comparable to those of standard GNNs. However, most real-world graphs are sparse. We therefore acknowledge as a limitation of our model that it does not scale to graphs with a very large number of nodes (e.g., hundreds of thousands or millions), since storing $\exp(\mathbf{A})$ in memory is infeasible for such graphs. Nevertheless, we should stress that the matrix product $\exp(\mathbf{A})\mathbf{M}$ (with $\mathbf{M}$ an arbitrary matrix) can be efficiently approximated as discussed in Appendix B.

## 4. Experimental Evaluation

In this section, we evaluate the INVGNN model on standard graph classification datasets and validate its expressiveness. We also demonstrate its effectiveness in outlier detection and node feature generation, and its interpretable nature.

### 4.1. Graph Classification

**Datasets.** We evaluate the proposed model on six datasets contained in the TUDataset collection (Morris et al., 2020): MUTAG, NCI1, PROTEINS, ENZYMES, IMDB-BINARY and IMDB-MULTI. We also evaluate the proposed model on ogbg-molhiv, a molecular property prediction dataset from the Open Graph Benchmark (OGB) (Hu et al., 2020).

**Baselines.** For the six datasets from the TUDataset collection, we compare the proposed model against the following nine GNN models: (1) DGCNN (Zhang et al., 2018); (2) DiffPool (Ying et al., 2018); (3) ECC (Simonovsky & Komodakis, 2017); (4) GIN (Xu et al., 2019); (5) GraphSAGE (Hamilton et al., 2017); (6) 3-step RWNN (Nikolentzos & Vazirgiannis, 2020); (7) $\pi$-GNN (Nikolentzos et al., 2022); (8) SPN (Abboud et al., 2022); and (9) Nested GNN (Zhang & Li, 2021). For the ogbg-molhiv, we compare the proposed model against the following eight GNN models: (1) GCN (Kipf & Welling, 2017); (2) GIN (Xu et al., 2019); (3) GCN-FLAG (Kong et al., 2020); (4) GIN-FLAG (Kong et al., 2020); (5) GSN (Bouritsas et al., 2022); (6) $\pi$-GNN (Nikolentzos et al., 2022); (7) ESAN (Bevilacqua et al., 2022); and (8) E-SPN (Abboud et al., 2022). For all baselines, we use the accuracies and ROC-AUC scores that are reported in prior work.

*Table 1.* Classification accuracy ($\pm$ standard deviation) of the proposed INVGNN model and the baselines on the datasets from the TUDataset collection. Best performance is highlighted in **bold**. NA means not available.

| | MUTAG | PROTEINS | NCI1 | ENZYMES | IMDB BINARY | IMDB MULTI |
|---|---|---|---|---|---|---|
| DGCNN | $84.0 \pm 6.7$ | $72.9 \pm 3.5$ | $76.4 \pm 1.7$ | $38.9 \pm 5.7$ | $69.2 \pm 3.0$ | $45.6 \pm 3.4$ |
| DiffPool | $79.8 \pm 7.1$ | $73.7 \pm 3.5$ | $76.9 \pm 1.9$ | $59.5 \pm 5.6$ | $68.4 \pm 3.3$ | $45.6 \pm 3.4$ |
| ECC | $75.4 \pm 6.2$ | $72.3 \pm 3.4$ | $76.2 \pm 1.4$ | $29.5 \pm 8.2$ | $67.7 \pm 2.8$ | $43.5 \pm 3.1$ |
| GIN | $84.7 \pm 6.7$ | $73.3 \pm 4.0$ | $\mathbf{80.0} \pm 1.4$ | $59.6 \pm 4.5$ | $71.2 \pm 3.9$ | $\mathbf{48.5} \pm 3.3$ |
| GraphSAGE | $83.6 \pm 9.6$ | $73.0 \pm 4.5$ | $76.0 \pm 1.8$ | $58.2 \pm 6.0$ | $68.8 \pm 4.5$ | $47.6 \pm 3.5$ |
| 3-step RWNN | $\mathbf{88.6} \pm 4.1$ | $74.3 \pm 3.3$ | $73.9 \pm 1.3$ | $57.6 \pm 6.3$ | $70.7 \pm 3.9$ | $47.8 \pm 3.5$ |
| $\pi$-GNN | $86.1 \pm 8.4$ | $73.6 \pm 3.5$ | $76.0 \pm 1.7$ | $60.3 \pm 4.1$ | $70.4 \pm 3.0$ | $\mathbf{48.5} \pm 3.5$ |
| SPN ($K = 1$) | NA | $71.0 \pm 3.7$ | $\mathbf{80.0} \pm 1.5$ | $67.5 \pm 5.5$ | NA | NA |
| SPN ($K = 5$) | NA | $74.2 \pm 2.7$ | $78.6 \pm 3.8$ | $\mathbf{69.4} \pm 6.2$ | NA | NA |
| Nested GNN | NA | $74.2 \pm 3.7$ | NA | $31.2 \pm 6.7$ | NA | NA |
| INVGNN | $84.9 \pm 5.4$ | $\mathbf{75.1} \pm 2.9$ | $76.4 \pm 2.4$ | $61.2 \pm 3.8$ | $\mathbf{71.5} \pm 4.8$ | $\mathbf{48.5} \pm 4.3$ |

Note that achieving state-of-the-art performance on these well-studied datasets is not the primary objective of this work. Given that several existing models are more expressive than INVGNN (Morris et al., 2019; Maron et al., 2019), we do not anticipate outperforming such methods. Instead, the purpose of the graph classification experiments is to demonstrate that the proposed model can achieve performance comparable to that of established GNN architectures, and that its invertibility does not negatively affect its empirical performance on real-world datasets. For this reason, we do not include comparisons with architectures that are more expressive and computationally more complex than the proposed model.

**Experimental setup.** Following Errica et al. (2020), we evaluate the model on TUDatasets using a 10-fold cross validation using their provided data splits. For all datasets, we set the batch size to 64 and the number of epochs to 500, using early stopping with a patience parameter of 50. We optimize the model with the Adam optimizer and a learning rate of 0.001. The $\exp(\mathbf{A})$ matrix of each graph is precomputed, and for stability, each entry of $\mathbf{A}$ is divided by the average of the largest eigenvalue across all graphs in the training set of each split. The hyperparameters tuned for each dataset are the hidden dimension size $\in \{32, 64, 128\}$, and the number of layers $\in \{2, 3, 4\}$. The sigmoid function is applied to the output of the hidden layers.

For the ogbg-molhiv dataset, we used the available predefined splits. We set the batch size to 128 and the number of epochs to 200. The hidden dimension size was chosen from $\{128, 256\}$, and the number of layers from $\{2, 3\}$. The rest of the experimental setup is the same as described above. All reported results are averaged over 10 runs.

**Results.** Table 1 illustrates the classification accuracy achieved by the proposed INVGNN model and the base-

*Table 2.* ROC-AUC score ($\pm$ standard deviation) of the different methods on the ogbg-molhiv dataset. Best performance is highlighted in **bold**.

| | ogbg-molhiv |
|---|---|
| GCN | $76.06 \pm 0.97$ |
| GIN | $75.58 \pm 1.40$ |
| GCN+FLAG | $76.83 \pm 1.02$ |
| GIN+FLAG | $76.54 \pm 1.14$ |
| GSN | $77.99 \pm 1.00$ |
| $\pi$-GNN | $79.12 \pm 1.50$ |
| ESAN | $78.00 \pm 1.42$ |
| E-SPN | $77.10 \pm 1.20$ |
| INVGNN | $77.61 \pm 1.61$ |

lines on the six datasets from the TUDataset collection. We observe that INVGNN is the best-performing model on three of the six datasets. On most datasets, the accuracy of our model is comparable to that of GIN, which can also match the expressive power of 1-WL. Notably, on PROTEINS, INVGNN provides an absolute improvement of $1.8\%$ in accuracy over GIN. However, on NCI1, GIN outperforms the proposed model by $3.6\%$. Overall, our results indicate that INVGNN achieves high performance on the six datasets from the TUDatasets collection, and that invertibility does not compromise its predictive ability.

Table 2 shows the ROC-AUC of the different methods on the ogbg-molhiv dataset. Even though INVGNN is not the best-performing method, it outperforms the GIN model by a substantial margin ($2.03\%$ absolute increase in ROC-AUC score). This experiment demonstrates the effectiveness of the proposed model on large graph classification datasets, as ogbg-molhiv is significantly larger (41,127 samples in total) than the considered datasets from the TUDatasest collection.

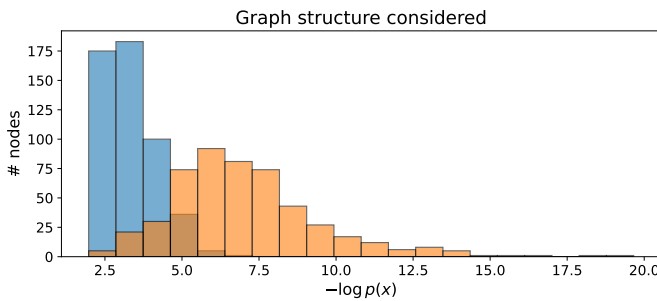 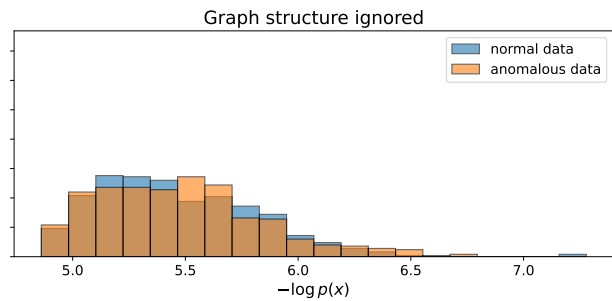

*Figure 2.* Histograms of the negative log-likelihoods of nodes from 20 graphs. Half of the graphs are Erdös-Rényi graphs constructed using the same procedure as those on which the model was trained (normal data). The remaining graphs exhibit a community structure and are considered outliers (anomalous data). In the right panel, the graph topology is ignored.

*Table 3.* Number of failures of the INVGNN model in distinguishing structurally different nodes that are distinguished by 1-WL, with varying hidden dimension and activation.

| Dim | Analytic | | | PwL | |
|---|---|---|---|---|---|
| | Tanh | Sigmoid | SiLU | ReLU | LeakyReLU |
| 1 | 2 | 2 | 4 | 5,583,565 | 28 |
| 5 | 0 | 0 | 0 | 25 | 24 |
| 10 | 0 | 0 | 0 | 21 | 19 |
| 20 | 0 | 0 | 0 | 16 | 14 |
| 57 | 0 | 0 | 0 | 14 | 12 |

## 4.2. Expressive Power

We next provide empirical validation of our theoretical results on the expressive power of the proposed model. We perform four iterations of the 1-WL algorithm on the 188 graphs of the MUTAG dataset. The number of nodes of the graphs range from 10 nodes to 28 nodes. The total number of nodes is equal to 3,371. The 1-WL algorithm assigns some color to each one of those 3,371 nodes. We then randomly initialize the parameters of INVGNN (we use four neighborhood aggregation layers to match the iterations of 1-WL), we perform a forward pass and we count the pairs of nodes that are assigned different colors by 1-WL, but the same vector representation by INVGNN. Note that the number of pairs of nodes with different colors is 5,583,565. We experiment with different analytic and piecewise linear functions and different hidden dimension sizes. Since the largest graph consists of 28 nodes, Theorem 3.2 suggests a dimension size of $2 \cdot 28 + 1 = 57$ to achieve expressive power equivalent to 1-WL for almost any values of parameters.

The results are illustrated in Table 3. We observe that, in the case of analytic activation functions, while our theoretical result suggests a dimension size of 57 to achieve expressive power equivalent to 1-WL for almost any choice of parameters, this level of expressiveness is already achieved with a dimension size as low as 5. On the other hand, the two piecewise linear functions fail to distinguish some structurally

different nodes, even with a dimension size of 57. For a dimension size of 1, the ReLU function appears unable to separate structurally different nodes, which is likely due to the model embedding all nodes into the same real value.

## 4.3. Outlier Detection

### 4.3.1. SYNTHETIC DATASET

Invertible models are commonly used for outlier detection. We next investigate whether INVGNN can detect nodes that are considered outliers. We construct a synthetic dataset consisting of 100 Erdös-Rényi graphs, each with 50 nodes and edge probability 0.27. Each node is annotated with a two-dimensional feature vector. For half of the nodes, the feature vectors are sampled from $\mathcal{N}(\mathbf{1}, 0.5\mathbf{I})$, while for the remaining nodes they are sampled from $\mathcal{N}(-\mathbf{1}, 0.5\mathbf{I})$. The INVGNN model is trained on this dataset using maximum likelihood estimation. We use 2 layers and train the model for 1,000 epochs. For stability, each entry of $\mathbf{A}$ is divided by the average of the largest eigenvalue across all graphs in the training set. After training, we can compute the log-likelihood of each node using the change-of-variables formula (Equation (1)). After training, we generate 10 graphs using the same procedure described above. In addition, we generate 10 graphs that exhibit a community structure with two equally sized communities and have a similar density to the Erdös–Rényi graphs. Nodes in the first community are annotated with feature vectors sampled from $\mathcal{N}(\mathbf{1}, 0.5\mathbf{I})$, while nodes in the second community are annotated with feature vectors sampled from $\mathcal{N}(-\mathbf{1}, 0.5\mathbf{I})$. We consider those 10 graphs as outliers. We feed all 20 graphs to the model and visualize the negative log-likelihoods of the nodes.

The results are illustrated in Figure 2. The left panel considers the graph structure, while the right panel treats node representations as independent, without interactions between them. We observe that the INVGNN model assigns lower negative log-likelihoods to nodes belonging to graphs drawn from the same distribution as the training set. On the other hand, nodes from graphs that exhibit a community structure

are assigned higher negative log-likelihoods. This indicates that the model assigns lower probability to such nodes and therefore identifies them as potential outliers. As expected, the model that ignores graph structure fails to distinguish outlier nodes (results shown in the right panel), since an equal number of nodes in both types of graphs are annotated with features drawn from the same distributions. This demonstrates that the proposed model can detect outliers in setting where models that treat graphs as unordered multisets of node representations fail.

### 4.3.2. REAL-WORLD DATASETS

Next, we evaluate the proposed model on five real-world datasets containing organically occurring outliers, namely Weibo (Zhao et al., 2020), Reddit (Kumar et al., 2019), Disney (Sánchez et al., 2013), Books (Sánchez et al., 2013) and Enron (Sánchez et al., 2013). The objective is to detect the outlier nodes. The proportion of outlier nodes in these datasets is $10.3\%$, $3.3\%$, $4.8\%$, $2.0\%$ and $0.04\%$, respectively. As before, INVGNN is trained on each dataset using maximum likelihood estimation. After training, the log-likelihood of each node is computed using the change-of-variables formula (Equation (1)). These log-likelihood values are then used as anomaly scores to evaluate outlier detection performance in terms of ROC-AUC. Note that in real-world unsupervised outlier detection settings, hyperparameter tuning and model selection are particularly challenging due to the absence of ground-truth labels (Liu et al., 2022). Following Liu et al. (2022), we evaluate multiple hyperparameter configurations and report the best achieved performance. In particular, we tune only the number of layers and the number of training epochs. Moreover, for stability, each entry of $\mathbf{A}$ is divided by its largest eigenvalue.

We compare the proposed method against six representative baseline approaches, including both standard and GNN-based outlier detection methods: (1) SCAN (Xu et al., 2007); (2) Radar (Li et al., 2017); (3) DOMINANT (Ding et al., 2019); (4) DONE (Bandyopadhyay et al., 2020); (5) AdONE (Bandyopadhyay et al., 2020); and (6) GAAN (Chen et al., 2020). For all baselines, we use ROC-AUC scores that are reported in prior work Liu et al. (2022).

The ROC-AUC scores achieved by the proposed model and the baseline methods are reported in Table 4. INVGNN achieves the best performance on the Disney dataset, while ranking second-best and third-best on the Books and Reddit datasets, respectively. Overall, the results demonstrate that the proposed approach is competitive with methods specifically designed for outlier detection. These findings further indicate that the model is able to effectively capture the underlying probability distribution of graph nodes, assigning lower log-likelihoods to outlier nodes.

*Table 4.* ROC-AUC performance of the proposed INVGNN model and the baselines on the five outlier detection datasets. For each method, we report the maximum performance over all hyperparameter configurations. Best performance is highlighted in **bold**.

|  | **Weibo** | **Reddit** | **Disney** | **Books** | **Enron** |
|---|---|---|---|---|---|
| SCAN | 70.8 | 50.0 | 56.1 | 52.4 | 58.1 |
| Radar | **99.0** | 56.9 | 51.8 | 52.8 | 80.8 |
| DOMINANT | 92.5 | 56.4 | 54.9 | 58.1 | **85.0** |
| DONE | 88.7 | **59.7** | 50.6 | 52.6 | 67.1 |
| AdONE | 87.6 | 58.1 | 59.2 | 56.1 | 53.6 |
| GAAN | 92.5 | 56.0 | 48.0 | **61.9** | 73.1 |
| INVGNN | 88.9 | 57.2 | **65.7** | 59.2 | 72.0 |

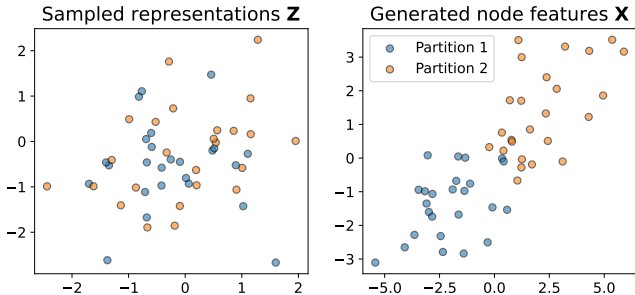

*Figure 3.* Visualization of the 50 vectors sampled from $\mathcal{N}(\mathbf{0}, \mathbf{I})$ and then fed into the INVGNN model with the inverse of the exponential adjacency matrix, and the resulting node features produced via the inverse mapping. Colors indicate partition memberships.

### 4.4. Node Feature Generation

We next investigate whether the INVGNN model can generate node features similar to those of the graphs on which the model is trained. We construct a synthetic dataset that comprises of 100 bipartite graphs. Each graph contains 50 nodes, equally divided between the two partitions (i.e., 25 nodes per partition). The probability of an edge existing between a node in the first partition and a node in the second partition is set to $0.3$. The nodes are annotated with two-dimensional feature vectors. Features of nodes in the first partition are drawn from $\mathcal{N}(\mathbf{1}, 0.5\mathbf{I})$, while those of nodes in the second partition from $\mathcal{N}(-\mathbf{1}, 0.5\mathbf{I})$. The INVGNN model is trained on this dataset using maximum likelihood estimation. We use 2 layers and train the model for 200 epochs. After training, we generate a random bipartite graph following the same procedure described above. We compute $\left(\exp(\mathbf{A})\right)^{-1}$ where $\mathbf{A}$ is the adjacency matrix of the generated bipartite graph. We then sample 50 two-dimensional vectors from $\mathcal{N}(\mathbf{0}, \mathbf{I})$, and feed both the inverse of the matrix exponential and the sampled vectors to the INVGNN model to perform the inverse mapping. The model then outputs features for the nodes.

The sampled vectors and the generated node features are

*Table 5.* Features of misclassified nodes in the Cora test set and of their neighbors that changed the most when the model output was modified to correctly classify each node. Features are shown separately for each ground-truth class. For the nodes themselves, we report the features with the largest increase, while for their neighbors we report those with the largest decrease.

| | Reinforcement_Learning | Theory | Case_Based | Neural_Networks | Probabilistic_Methods | Rule_Learning | Genetic_Algorithms |
|---|---|---|---|---|---|---|---|
| Node (Largest ↑) | converg, limit, issu, complet, heurist, theoret, abstract, program, lead, applic | limit, issu, complet, approxim, theoret, prove, lead, understand, demonstr, converg | limit, issu, heurist, improv, situat, complet, approxim, theoret, context, demonstr | issu, limit, heurist, converg, complet, theoret, context, approxim, situat, recognit | level, improv, properti, sequenc, genet, variabl, call, rate, architectur, featur | properti, level, genet, sequenc, call, improv, activ, class, simul, logic | level, genet, nois, consist, finit, program, properti, result, scale, capabl |
| Neighbors (Largest ↓) | converg, issu, theoret, complet, limit, heurist, program, abstract, lead, reason | limit, issu, approxim, complet, understand, theoret, prove, lead, applic, converg | limit, issu, heurist, complet, approxim, theoret, situat, context, improv, solut | issu, heurist, complet, converg, limit, theoret, context, approxim, recognit, error | level, improv, properti, sequenc, genet, variabl, factor, suggest, call, architectur | properti, level, genet, sequenc, activ, simul, class, solv, logic, call | program, analyz, finit, activ, nois, control, reason, consist, independ, understand |

illustrated in Figure 3. We observe that the model successfully produces two distinct groups of features: one group predominantly corresponding to nodes from the first partition and the other to nodes from the second partition. In addition, the two groups are well aligned with the distributions from which the training node features were drawn, namely $\mathcal{N}(\mathbf{1}, 0.5\mathbf{I})$ and $\mathcal{N}(-\mathbf{1}, 0.5\mathbf{I})$.

## 4.5. Decision Explanation

We next use the INVGNN model to investigate how the initial node features would need to change for the model to correctly classify previously misclassified samples. We experiment with the Cora dataset (Yang et al., 2016). Since the publicly available version of the dataset does not provide a vocabulary, we recomputed bag-of-words features for all nodes. To do so, we retrieved the abstracts of all papers, lowercased all words, removed stopwords and tokens containing numeric characters, applied stemming, and discarded terms appearing in fewer than 100 abstracts. Following these preprocessing steps, the resulting vocabulary consists of 268 terms. We used 2,068 nodes for training and the remaining 640 nodes for testing. The model consists of 2 layers, each of dimension 268, to allow end-to-end invertibility. Although the model produces a 268-dimensional vector for each node, it effectively assigns the highest probabilities to the first 7 components, corresponding to the actual classes. We train the model for 50 epochs with a learning rate of 0.001. For stability, each entry of $\mathbf{A}$ is divided by its largest eigenvalue.

Once the model is trained (it achieved a classification accuracy of 82.14% on the test set), we identify misclassified nodes of each class in the test set. For each such node, we increase the logit of the correct class so that it surpasses the largest logit by a small constant, and then perform the inverse mapping. In other words, we modify the model's predictions so that each previously misclassified node is correctly classified. We then analyze the resulting changes in the node's initial features, as well as in the initial features of its neighbors. This analysis provides insight into which aspects of the initial features would need to be dif-

ferent in order for the model not to misclassify these nodes. We provide in Table 5 the features that changed the most for misclassified nodes of each class. For the nodes themselves, we list the features with the largest increase, while for their neighbors we list those with the largest decrease. The presence of certain terms in specific classes is not surprising, such as "genet" in Genetic_Algorithms, "variabl" in Probabilistic_Methods, and "recognit" in Neural_Networks. Papers whose abstracts contain these terms are expected to belong to the corresponding classes. It is worth to mention that for each class, the features that increase the most for the nodes largely coincide with those that decrease the most in their neighboring nodes. Since Cora is a homophilic dataset, the majority of the neighbors of a given node contain useful information for classifying that node. Intuitively, the observed behavior can be interpreted as a flow of information: to change the prediction for a node, the model diffuses the node's uninformative or confusing features to its neighbors while aggregating more discriminative information from them.

## 4.6. Time and Memory Complexity Analysis

As discussed above, one layer of the proposed model has time complexity $\mathcal{O}(n^2)$, since the matrix exponential $\exp(\mathbf{A})$ is dense. The memory complexity also scales as $\mathcal{O}(n^2)$ since the matrix exponential is precomputed and stored in memory. To illustrate the scalability of the INvGNN model, we constructed synthetic node classification datasets in which graphs are generated using a stochastic block model. For each graph, we measured the model's runtime and memory usage, as well as the time required to compute $\exp(\mathbf{A})$ from the graph's adjacency matrix $\mathbf{A}$. Specifically, we generated graphs with equally sized blocks and set the intra-block edge probability to 0.4 and the inter-block edge probability to 0.05. Nodes in the first block were assigned to class 0, while nodes in the second block were assigned to class 1. We set the hidden dimension size equal to 64 and the number of neighborhood aggregation layers to 3. The output of the last neighborhood aggregation layer was fed to a fully-connected layer to produce class probabil-

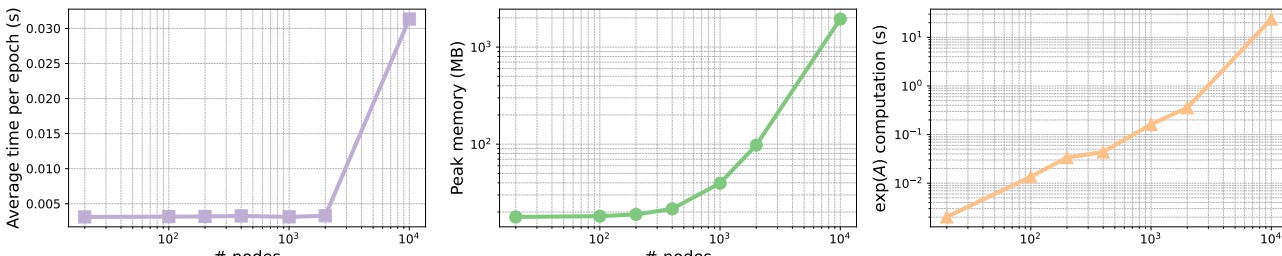

*Figure 4.* Complexity analysis of the INVGNN model trained on graphs of increasing size. Left: average running time per epoch. Middle: peak memory consumption. Right: computation of the matrix exponential of the adjacency matrix.

ities. The model was trained for 50 epochs by minimizing the cross-entropy loss. All experiments were conducted on a machine equipped with an Intel Xeon CPU running at 2.20 GHz (2 cores, 16 GB RAM) and an NVIDIA Tesla T4 GPU with 16 GB of memory.

The results are provided in Figure 4. The average time per training epoch of INVGNN remains approximately constant for graphs with up to 2,000 nodes. For the graph containing 10,000 nodes, it increases by approximately one order of magnitude. Nevertheless, a runtime of approximately 0.03 seconds per epoch remains well within acceptable limits. We can also see that the memory requirements of the model increase with graph size. While less than 22 MB are required for graphs up to 400 nodes, the memory usage increases to apprximately 40, 97 and 1,945 MB for graphs with 1,000, 2,000 and 10,000 nodes, respectively. Note that computing the matrix exponential $\exp(\mathbf{A})$ requires an eigenvalue decomposition of the adjacency matrix $\mathbf{A}$, which has a time complexity of $\mathcal{O}(n^3)$. Figure 4 shows that $\exp(\mathbf{A})$ can be computed in less than one second for graphs with up to 2,000 nodes, and requires approximately 23 seconds for a graph with 10,000 nodes. Although this computation introduces an overhead that standard GNNs do not incur, it is performed only once, and the resulting increase in total runtime is not dramatic.

## 5. Conclusion

In this paper, we introduced an invertible GNN layer that enables the construction of fully invertible GNN models, which we refer to as INVGNNs. Invertibility makes these models particularly suitable for tasks such as density estimation and data generation. Our proposed layer leverages the matrix exponential of the adjacency matrix, ensuring that the transformation is always invertible. We demonstrated that stacking instances of this layer in sequence allows the resulting models to match the expressive power of the 1-WL algorithm, thus providing both invertibility and sufficient expressiveness. The INVGNN model was evaluated on graph classification tasks, achieving performance comparable to well-established GNN architectures such as GIN. Addition-

ally, we empirically showed that INVGNN can detect outlier nodes, generate node features and provide explanations of GNN decisions.

## Acknowledgements

We thank the anonymous reviewers for their constructive feedback. N.N. is supported by the NOMIS Foundation. GCP resources were provided by the National Infrastructures for Research and Technology GRNET and funded by the EU Recovery and Resiliency Facility.

## Impact Statement

This paper presents work whose goal is to advance the field of Machine Learning. There are many potential societal consequences of our work, none which we feel must be specifically highlighted here.

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

# A. Proof of Theorem 3.2

Before proceeding with the proof of the Theorem, we establish three Lemmas that will be used in the proof. If the multisets of nonzero entries in the rows of the matrix exponentials of the adjacency matrix corresponding to two nodes are different, then the nodes are structurally distinct. The purpose of the Lemmas is to show that if the multisets are equal, then the entries of the rows that correspond to the two nodes are identical and uniquely determined. Moreover, the multisets of entries corresponding to their direct neighbors are also equal, and these entries are distinct from those associated with nodes that are not direct neighbors. Therefore, the node itself and its neighbors can be uniquely identified.

We will first show that if the multisets of nonzero entries in row $\exp(\mathbf{A})_{i,:}$ and row $\exp(\mathbf{A})'_{j,:}$ are equal, then $\exp(\mathbf{A})_{i,i} = \exp(\mathbf{A}')_{j,j}$.

**Lemma A.1.** *Let $G = (V, E)$, $G' = (V', E')$ denote two unweighted, undirected graphs on $n$ nodes, and let $\mathbf{A}$, $\mathbf{A}'$ denote their respective adjacency matrices. In addition, let $v_i \in V$ and $v'_j \in V'$. If the multiset of nonzero entries in row $\exp(\mathbf{A})_{i,:}$ is equal to the multiset of nonzero entries in row $\exp(\mathbf{A}')_{j,:}$, then $\exp(\mathbf{A})_{i,i} = \exp(\mathbf{A})_{j,j}$.*

*Proof.* Because $\mathbf{A}$ and $\mathbf{A}'$ are real symmetric matrices, they admit spectral decompositions. Let $\lambda_1, \ldots, \lambda_n$ denote the eigenvalues of matrix $\mathbf{A}$ and let $\mathbf{U}$ be its orthogonal matrix of eigenvectors. Likewise, let $\lambda'_1, \ldots, \lambda'_n$ and $\mathbf{U}'$ denote the eigenvalues and eigenvectors of $\mathbf{A}'$. Since the multiset of entries in row $\exp(\mathbf{A})_{i,:}$ is equal to the multiset of entries in row $\exp(\mathbf{A}')_{j,:}$, we have:

$$\exp(\mathbf{A})_{i,:} \, \exp(\mathbf{A})_{i,:}^{\top} = \exp(\mathbf{A}')_{j,:} \, \exp(\mathbf{A}')_{j,:}^{\top}$$
$$\implies \exp(\mathbf{A})_{i,i}^2 = \exp(\mathbf{A}')_{j,j}^2$$
$$\implies \sum_{k=1}^{n} (e^{\lambda_k})^2 \mathbf{U}_{i,k}^2 = \sum_{k=1}^{n} (e^{\lambda'_k})^2 (\mathbf{U}'_{i,k})^2$$
$$\implies \sum_{k=1}^{n} e^{2\lambda_k} \mathbf{U}_{i,k}^2 = \sum_{k=1}^{n} e^{2\lambda'_k} (\mathbf{U}'_{i,k})^2$$
$$\implies \sum_{k=1}^{n} e^{2\lambda_k} \mathbf{U}_{i,k}^2 - \sum_{k=1}^{n} e^{2\lambda'_k} (\mathbf{U}'_{i,k})^2 = 0$$

Let $\mathcal{M} = \{\mu_1, \mu_2, \ldots, \mu_d\}$ be the set of all distinct eigenvalues present in either $\mathbf{A}$ or $\mathbf{A}'$. We can group the terms of the equation above by these distinct exponents to form a linear combination:

$$\sum_{m=1}^{d} c_m e^{2\mu_m} = 0$$

where each coefficient $c_m$ is equal to the difference between the sum of $\mathbf{U}_{i,k}^2$ terms (for all $k$ where $\lambda_k = \mu_m$) and the sum of $(\mathbf{U}'_{i,k})^2$ terms (for all $k$ where $\lambda'_k = \mu_m$). Because the elements of $\mathbf{A}$ and $\mathbf{A}'$ are integers, their eigenvalues and the entries of their corresponding eigenvectors are algebraic numbers. The algebraic numbers form a field, and they are closed under addition, subtraction, and multiplication. Therefore, every coefficient $c_m$ is an algebraic number.

According to the Lindemann–Weierstrass theorem (Baker, 2022) (Thm. 1.4), if $\mu_1, \ldots, \mu_d$ are distinct algebraic numbers, then their exponentials $e^{\mu_1}, \ldots, e^{\mu_d}$ are linearly independent over the algebraic numbers. Therefore, the linear combination equal to zero requires that:

$$c_1 = c_2 = \cdots = c_d = 0$$

Since every coefficient $c_m$ is exactly zero, any linear combination scaled by these coefficients is also zero. Therefore, we have:

$$\sum_{m=1}^{d} c_m e^{\mu_m} = 0$$

Expanding $c_m$ back into its constituent eigenvector products, we have:

$$\sum_{k=1}^{n} e^{\lambda_k} \mathbf{U}_{i,k}^2 - \sum_{k=1}^{n} e^{\lambda'_k} (\mathbf{U}'_{i,k})^2 = 0$$

$$\implies \sum_{k=1}^{n} e^{\lambda_k} \mathbf{U}_{i,k}^2 = \sum_{k=1}^{n} e^{\lambda_k'} (\mathbf{U}_{i,k}')^2$$

$$\implies \exp(\mathbf{A})_{i,i} = \exp(\mathbf{A})_{j,j}'$$

which concludes the proof. $\qquad\square$

If the multisets of nonzero entries in row $\exp(\mathbf{A})_{i,:}$ and row $\exp(\mathbf{A})_{j,:}'$ are equal, then there exists a permutation $\pi : [n] \to [n]$ of the nodes of $G'$ with $\pi(j) = i$ such that $\mathbf{A}_{i,:} = [\mathbf{A}_\pi']_{i,:}$. Here, we have assumed that both graphs consist of $n$ nodes. If a graph contains fewer than $n$ nodes, we can always pad its adjacency matrix with all-zero rows and columns to increase the number of nodes to $n$. The next Lemma assumes that the aforementioned permutation has been applied to reorder the nodes of $G'$. It states that if the rows $\exp(\mathbf{A})_{i,:}$ and $\exp(\mathbf{A})_{j,:}'$ have equal multisets of nonzero entries, then the corresponding rows of any power of the adjacency matrices of $G$ and $G'$ also have equal multisets of nonzero entries.

**Lemma A.2.** *Let $G = (V, E), G' = (V', E')$ denote two unweighted, undirected graphs and let $\mathbf{A}, \mathbf{A}'$ denote their respective adjacency matrices. If the $i$-th rows of their matrix exponentials are equal, such that $\exp(\mathbf{A})_{i,:} = \exp(\mathbf{A}')_{i,:}$, then $\mathbf{A}_{i,:}^r = (\mathbf{A}')_{i,:}^r$ for any $r \in \mathbb{N} \cup \{0\}$.*

*Proof.* Because $\mathbf{A}$ and $\mathbf{A}'$ are real symmetric matrices, they admit spectral decompositions. Let $\lambda_1, \ldots, \lambda_n$ denote the eigenvalues of matrix $\mathbf{A}$ and let $\mathbf{U}$ be its orthogonal matrix of eigenvectors. Likewise, let $\lambda_1', \ldots, \lambda_n'$ and $\mathbf{U}'$ denote the eigenvalues and eigenvectors of $\mathbf{A}'$. For any $j \in [n]$, we have:

$$\exp(\mathbf{A})_{i,j} = \exp(\mathbf{A}')_{i,j}$$

$$\implies \sum_{k=1}^{n} e^{\lambda_k} \mathbf{U}_{i,k} \mathbf{U}_{j,k} = \sum_{k=1}^{n} e^{\lambda_k'} \mathbf{U}_{i,k}' \mathbf{U}_{j,k}'$$

$$\implies \sum_{k=1}^{n} e^{\lambda_k} \mathbf{U}_{i,k} \mathbf{U}_{j,k} - \sum_{k=1}^{n} e^{\lambda_k'} \mathbf{U}_{i,k}' \mathbf{U}_{j,k}' = 0$$

Let $\mathcal{M} = \{\mu_1, \mu_2, \ldots, \mu_d\}$ be the set of all distinct eigenvalues present in either $\mathbf{A}$ or $\mathbf{A}'$. We can group the terms of the equation above by these distinct exponents to form a linear combination:

$$\sum_{m=1}^{d} c_m e^{\mu_m} = 0$$

where each coefficient $c_m$ is equal to the difference between the sum of $\mathbf{U}_{i,k} \mathbf{U}_{j,k}$ terms (for all $k$ where $\lambda_k = \mu_m$) and the sum of $\mathbf{U}_{i,k}' \mathbf{U}_{j,k}'$ terms (for all $k$ where $\lambda_k' = \mu_m$). Because the elements of $\mathbf{A}$ and $\mathbf{A}'$ are integers, their eigenvalues and the entries of their corresponding eigenvectors are algebraic numbers. The algebraic numbers form a field, and they are closed under addition, subtraction, and multiplication. Therefore, every coefficient $c_m$ is an algebraic number.

According to the Lindemann–Weierstrass theorem (Baker, 2022) (Thm. 1.4), if $\mu_1, \ldots, \mu_d$ are distinct algebraic numbers, then their exponentials $e^{\mu_1}, \ldots, e^{\mu_d}$ are linearly independent over the algebraic numbers. Therefore, the linear combination equal to zero requires that:

$$c_1 = c_2 = \cdots = c_d = 0$$

Since every coefficient $c_m$ is exactly zero, any linear combination scaled by these coefficients is also zero. Specifically, for any $r \in \mathbb{N} \cup \{0\}$, we have:

$$\sum_{m=1}^{d} c_m \mu_m^r = 0$$

Expanding $c_m$ back into its constituent eigenvector products, we have:

$$\sum_{k=1}^{n} \lambda_k^r \mathbf{U}_{i,k} \mathbf{U}_{j,k} - \sum_{k=1}^{n} (\lambda_k')^r \mathbf{U}_{i,k}' \mathbf{U}_{j,k}' = 0$$

$$\implies \sum_{k=1}^{n} \lambda_k^r \mathbf{U}_{i,k} \mathbf{U}_{j,k} = \sum_{k=1}^{n} (\lambda_k')^r \mathbf{U}_{i,k}' \mathbf{U}_{j,k}'$$

$$\Longrightarrow \mathbf{A}_{i,j}^r = (\mathbf{A}')_{i,j}^r$$

Because this holds for every $j \in [n]$, it follows that the entire $i$-th rows are equal, concluding the proof. $\square$

From the above Lemma, we conclude that the multisets of entries of $\exp(\mathbf{A})_{i,:}$ and $\exp(\mathbf{A}')_{i,:}$ corresponding to the neighbors of nodes $v_i$ and $v_i'$ are equal.

The final Lemma states that if two entries in the same row of the matrix exponential of the adjacency matrix are equal, then the corresponding entries in every power of the adjacency matrix are also equal.

**Lemma A.3.** *Let* $G = (V, E)$ *denote an unweighted, undirected graph and* $\mathbf{A}$ *denote its adjacency matrix. If* $\exp(\mathbf{A})_{i,j} = \exp(\mathbf{A})_{i,\ell}$, *then* $\mathbf{A}_{i,j}^r = \mathbf{A}_{i,\ell}^r$ *for all* $r \in \mathbb{N} \cup \{0\}$.

*Proof.* Since $\mathbf{A}$ is a real symmetric matrix, it admits a spectral decomposition. Let $\lambda_1, \dots, \lambda_n$ denote its eigenvalues and let $\mathbf{U}$ be its orthogonal matrix of eigenvectors. Let $\mu_1, \dots, \mu_m$ denote the distinct eigenvalues. Let $I_i$ denote the the set of indices $j \in [n]$ such that $\lambda_j = \mu_i$, i.e., $I_i = \{j \in [n] \mid \lambda_j = \mu_i\}$. Then, we have that:

$$\exp(\mathbf{A})_{i,j} = \exp(\mathbf{A})_{i,\ell} \Longrightarrow \sum_{k=1}^{n} e^{\lambda_k} \mathbf{U}_{i,k} \mathbf{U}_{j,k} = \sum_{k=1}^{n} e^{\lambda_k} \mathbf{U}_{i,k} \mathbf{U}_{\ell,k}$$

$$\Longrightarrow \sum_{k=1}^{d} \left( \sum_{m \in I_k} \mathbf{U}_{i,m} \mathbf{U}_{j,m} \right) e^{\mu_k} - \sum_{k=1}^{d} \left( \sum_{m \in I_k} \mathbf{U}_{i,m} \mathbf{U}_{\ell,m} \right) e^{\mu_k} = 0 \quad \left( \begin{array}{l} \text{group terms that share the} \\ \text{same distinct eigenvalue} \end{array} \right)$$

$$\Longrightarrow \sum_{k=1}^{d} \left( \sum_{m \in I_k} \left( \mathbf{U}_{i,m} \mathbf{U}_{j,m} - \mathbf{U}_{i,m} \mathbf{U}_{\ell,m} \right) \right) e^{\mu_k} = 0$$

$$\Longrightarrow \sum_{m \in I_k} \left( \mathbf{U}_{i,m} \mathbf{U}_{j,m} - \mathbf{U}_{i,m} \mathbf{U}_{\ell,m} \right) = 0 \quad \text{for all } k \in [d] \quad \left( \begin{array}{c} \text{due to Lindemann-Weierstrass} \\ \text{Theorem} \end{array} \right)$$

$$\Longrightarrow \sum_{m \in I_k} \mathbf{U}_{i,m} \mathbf{U}_{j,m} = \sum_{m \in I_k} \mathbf{U}_{i,m} \mathbf{U}_{\ell,m} \quad \text{for all } k \in [d]$$

Then, for any $r \in \mathbb{N} \cup \{0\}$, we have that:

$$\mathbf{A}_{i,j}^r = \sum_{k=1}^{n} \lambda_k^r \mathbf{U}_{i,k} \mathbf{U}_{j,k} = \sum_{k=1}^{d} \left( \sum_{m \in I_k} \mathbf{U}_{i,m} \mathbf{U}_{j,m} \right) \mu_k^r = \sum_{k=1}^{d} \left( \sum_{m \in I_k} \mathbf{U}_{i,m} \mathbf{U}_{\ell,m} \right) \mu_k^r = \sum_{k=1}^{n} \lambda_k^r \mathbf{U}_{i,k} \mathbf{U}_{\ell,k} = \mathbf{A}_{i,\ell}^r$$

which concludes the proof. $\square$

The above Lemma implies that if two entries in a row of some power of $\mathbf{A}$ are different, then the corresponding entries of $\exp(\mathbf{A})$ are also different. Therefore, the entries of $\exp(\mathbf{A})_{i,:}$ associated with neighbors of node $v_i$ are distinct from the remaining entries because the corresponding entries of $\mathbf{A}$ (i.e., first power of $\mathbf{A}$) differ. Likewise, $\exp(\mathbf{A})_{i,i}$ differs from all the remaining entries of $\exp(\mathbf{A})_{i,:}$ since $\mathbf{A}_{i,i}^0 \neq \mathbf{A}_{i,j}^0$ for any $j \neq i$.

We also state the following Theorem by Amir et al. (2023) which we will use to show that our model can be as expressive as 1-WL.

**Theorem A.4** (Amir et al. (2023))**.** *Let* $\sigma : \mathbb{R} \to \mathbb{R}$ *be an analytic non-polynomial function. Let* $n, d \in \mathbb{N}$, *and set* $m = 2n + 1$. *Let also* $\Sigma \subset \mathbb{R}^d$ *be any infinite countable alphabet. Then for Lebesgue almost any* $\mathbf{A} \in \mathbb{R}^{m \times d}, \mathbf{b} \in \mathbb{R}^m$, *the function* $f : \mathcal{M}_{\leq n}(\Sigma) \to \mathbb{R}^m$ *given by*

$$f(\mu) = \sum_{i=1}^{n} w_i \sigma \left( \mathbf{A} \mathbf{x}_i + \mathbf{b} \right) \text{ for } \mu = \sum_{i=1}^{n} w_i \delta_{\mathbf{x}_i} \tag{3}$$

*is injective.*

We will use the above Theorem to show that, almost surely, different representations are learned for certain nodes. We repeat below, for convenience, the node update rule of the proposed INVGNN model:

$$\mathbf{h}_i^{(k)} = \sum_{j=1}^{n} \exp\left(\mathbf{A}\right)_{ij} f\left(\mathbf{h}_j^{(k-1)} \mathbf{W}^{(k)} + \mathbf{b}^{(k)}\right)$$

Note that the $w_1, \ldots, w_n$ values in the Theorem A.4 correspond to the $\exp\left(\mathbf{A}\right)_{i,1}, \ldots, \exp\left(\mathbf{A}\right)_{i,n}$ entries of a row of $\exp\left(\mathbf{A}\right)$, while the the $\mathbf{x}_1, \ldots, \mathbf{x}_n$ vectors correspond to the node representations $\mathbf{h}_1^{(k-1)}, \ldots, \mathbf{h}_n^{(k-1)}$. Let $v_i$ and $v_j'$ be nodes in graphs $G$ and $G'$, respectively, where $G, G' \in \mathcal{G}_{\leq n}(\Sigma)$. Let $f$ be an analytic non-polynomial function, and let the hidden dimension size be at least equal to $2n + 1$. The above Theorem implies that if the multisets $\{\!\!\{(\exp\left(\mathbf{A}\right)_{i,1}, \mathbf{h}_1^{(k-1)}), \ldots, (\exp\left(\mathbf{A}\right)_{i,n}, \mathbf{h}_n^{(k-1)})\}\!\!\}$ and $\{\!\!\{(\exp\left(\mathbf{A}'\right)_{j,1}, \mathbf{h}_1'^{(k-1)}), \ldots, (\exp\left(\mathbf{A}'\right)_{j,n'}, \mathbf{h}_{n'}'^{(k-1)})\}\!\!\}$ are not equal, then $\mathbf{h}_i^{(k)} \neq \mathbf{h}_j'^{(k)}$ for Lebesgue almost any $\mathbf{W}^{(k)}, \mathbf{b}^{(k)}$. We will use this in the proof of Theorem 3.2 below.

We now proceed with the proof of the Theorem. If two nodes are structurally identical, it is easy to show that the proposed model learns identical representations for them. We next prove that whenever the 1-WL algorithm assigns different colors to two nodes, the proposed model almost surely learns different representations for them.

Let $c^{(k)}(v)$ denote the color that 1-WL assigns to node $v$ after $k \in \mathbb{N}$ iterations. The color of a node at iteration $k$ of 1-WL encodes the previous color of the node and the multiset of previous colors of its neighbors:

$$c^{(k)}(v) = \mathrm{hash}\big(c^{(k-1)}(v), \{\!\!\{c^{(k-1)}(u) \mid u \in \mathcal{N}(v)\}\!\!\}\big)$$

where $\mathrm{hash}(\cdot)$ assigns a new compressed identifier to the tuple.

We will show that if $c^{(k)}(v_i) \neq c^{(k)}(v_j')$, then $\mathbf{h}_i^{(k)} \neq \mathbf{h}_j'^{(k)}$. We proceed by induction on $k$. For $k = 1$, if $c^{(0)}(v_i) \neq c^{(0)}(v_j')$, then also $\mathbf{h}_i^{(0)} \neq \mathbf{h}_j'^{(0)}$ holds.

For the induction step, suppose that the representations learned by the proposed model distinguish all node pairs distinguished by 1-WL after $k-1$ iterations. That is, if $c^{(k-1)}(v_i) \neq c^{(k-1)}(v_j')$, then $\mathbf{h}_i^{(k-1)} \neq \mathbf{h}_j'^{(k-1)}$.

Now suppose that for two nodes $v_i, v_j'$ (possibly from different graphs) we have that:

$$c^{(k)}(v_i) \neq c^{(k)}(v_j') \implies \mathrm{hash}\Big(c^{(k-1)}(v_i), \{\!\!\{c^{(k-1)}(v_\ell) \mid v_\ell \in \mathcal{N}(v_i)\}\!\!\}\Big) \neq \mathrm{hash}\Big(c^{(k-1)}(v_j'), \{\!\!\{c^{(k-1)}(v_\ell') \mid v_\ell' \in \mathcal{N}(v_j')\}\!\!\}\Big)$$

This can happen in one of the following three cases:

**Case (i)**: if $c^{(k-1)}(v_i) \neq c^{(k-1)}(v_j')$

If $\{\!\!\{\exp\left(\mathbf{A}\right)_{i,1}, \ldots, \exp\left(\mathbf{A}\right)_{i,n}\}\!\!\} \neq \{\!\!\{\exp\left(\mathbf{A}'\right)_{j,1}, \ldots, \exp\left(\mathbf{A}'\right)_{j,n'}\}\!\!\}$, then, by Theorem A.4, the two nodes obtain different representations. Suppose that $\{\!\!\{\exp\left(\mathbf{A}\right)_{i,1}, \ldots, \exp\left(\mathbf{A}\right)_{i,n}\}\!\!\} = \{\!\!\{\exp\left(\mathbf{A}'\right)_{j,1}, \ldots, \exp\left(\mathbf{A}'\right)_{j,n'}\}\!\!\}$. By Lemma A.1, the elements of the multisets corresponding to nodes $v_i$ and $v_j'$ are equal, i.e., $\exp\left(\mathbf{A}\right)_{i,i} = \exp\left(\mathbf{A}'\right)_{j,j}$. Furthermore, by Lemma A.3, these two elements are distinct from all other elements in the multisets because, in the 0-th power of the adjacency matrix, they are equal to 1, while all other elements are equal to 0. Therefore, $\exp\left(\mathbf{A}\right)_{i,i} \neq \exp\left(\mathbf{A}\right)_{i,\ell}$ for any $\ell \in [n] \setminus \{i\}$. Likewise, $\exp\left(\mathbf{A}'\right)_{j,j} \neq \exp\left(\mathbf{A}'\right)_{j,\ell}$ for any $\ell \in [n] \setminus \{j\}$. Since $\mathbf{h}_i^{(k-1)} \neq \mathbf{h}_j'^{(k-1)}$ by the induction hypothesis, the pairs $(\exp\left(\mathbf{A}\right)_{i,i}, \mathbf{h}_i^{(k-1)})$ and $(\exp\left(\mathbf{A}'\right)_{j,j}, \mathbf{h}_j'^{(k-1)})$ are different from each other, and Theorem A.4 implies that $\mathbf{h}_i^{(k)} \neq \mathbf{h}_j'^{(k)}$ holds for Lebesgue almost any parameter values.

**Case (ii)**: if $\{\!\!\{c^{(k-1)}(v_\ell) \mid v_\ell \in \mathcal{N}(v_i)\}\!\!\} \neq \{\!\!\{c^{(k-1)}(v_\ell') \mid v_\ell' \in \mathcal{N}(v_j')\}\!\!\}$

Once again, if $\{\!\!\{\exp\left(\mathbf{A}\right)_{i,1}, \ldots, \exp\left(\mathbf{A}\right)_{i,n}\}\!\!\} \neq \{\!\!\{\exp\left(\mathbf{A}'\right)_{j,1}, \ldots, \exp\left(\mathbf{A}'\right)_{j,n'}\}\!\!\}$, then, by Theorem A.4, the two nodes obtain different representations. Suppose that $\{\!\!\{\exp\left(\mathbf{A}\right)_{i,1}, \ldots, \exp\left(\mathbf{A}\right)_{i,n}\}\!\!\} = \{\!\!\{\exp\left(\mathbf{A}'\right)_{j,1}, \ldots, \exp\left(\mathbf{A}'\right)_{j,n'}\}\!\!\}$. By Lemma A.2, the elements in the multisets that correspond to neighbors of $v_i$ and $v_j'$ have identical values for all powers of the adjacency matrix and therefore also in the multisets of entries of the matrix exponentials. We thus have that $\{\!\!\{\exp\left(\mathbf{A}\right)_{i,\ell} \mid v_\ell \in \mathcal{N}(v_i)\}\!\!\} = \{\!\!\{\exp\left(\mathbf{A}'\right)_{j,\ell} \mid v_\ell' \in \mathcal{N}(v_j')\}\!\!\}$. Furthermore, by Lemma A.3, those elements are different from those of nodes that are not neighbors of $v_i$ and $v_j'$ since their entries in the adjacency matrix are equal to 1, while the entries

corresponding to non-neighbors are equal to 0. Therefore, $\{\!\{\exp\left(\mathbf{A}\right)_{i,\ell} \mid v_\ell \in \mathcal{N}(v_i)\}\!\} \cap \{\!\{\exp\left(\mathbf{A}\right)_{i,\ell} \mid v_\ell \notin \mathcal{N}(v_i)\}\!\} = \emptyset$. Likewise, $\{\!\{\exp\left(\mathbf{A}'\right)_{j,\ell} \mid v'_\ell \in \mathcal{N}(v'_j)\}\!\} \cap \{\!\{\exp\left(\mathbf{A}'\right)_{i,\ell} \mid v'_\ell \notin \mathcal{N}(v'_j)\}\!\} = \emptyset$. Therefore, we also have that $\{\!\{\exp\left(\mathbf{A}\right)_{i,\ell} \mid v_\ell \in \mathcal{N}(v_i)\}\!\} \cap \{\!\{\exp\left(\mathbf{A}'\right)_{i,\ell} \mid v'_\ell \notin \mathcal{N}(v'_j)\}\!\} = \emptyset$ and $\{\!\{\exp\left(\mathbf{A}'\right)_{j,\ell} \mid v'_\ell \in \mathcal{N}(v'_j)\}\!\} \cap \{\!\{\exp\left(\mathbf{A}\right)_{i,\ell} \mid v_\ell \notin \mathcal{N}(v_i)\}\!\} = \emptyset$. Since we have $\{\!\{\mathbf{h}_\ell^{(k-1)} \mid v_\ell \in \mathcal{N}(v_i)\}\!\} \neq \{\!\{\mathbf{h}_\ell'^{(k-1)} \mid v'_\ell \in \mathcal{N}(v'_j)\}\!\}$ by the induction hypothesis, the multisets of pairs $\{\!\{(\exp\left(\mathbf{A}\right)_{i,\ell}, \mathbf{h}_\ell^{(k-1)}) \mid v_\ell \in \mathcal{N}(v_i)\}\!\}$ and $\{\!\{(\exp\left(\mathbf{A}'\right)_{j,\ell}, \mathbf{h}_\ell'^{(k-1)}) \mid v'_\ell \in \mathcal{N}(v'_j)\}\!\}$ are different from each other, and Theorem A.4 implies that $\mathbf{h}_i^{(k)} \neq \mathbf{h}_j'^{(k)}$ holds for Lebesgue almost any parameter values.

**Case (iii)**: if both above conditions hold.
Based on the above, $\mathbf{h}_i^{(k)} \neq \mathbf{h}_j'^{(k)}$ holds for Lebesgue almost any parameter values.

## B. Sparse Chebyshev Approximation of $\exp(\mathbf{A})$

Since we have assumed undirected graphs, the adjacency matrix $\mathbf{A}$ is symmetric. Most real-world graphs are sparse, and therefore the associated matrix $\mathbf{A}$ is also sparse. We next present how we can compute the action of the exponential operator

$$\mathbf{Y} = \exp(\mathbf{A})\,\mathbf{M}$$

for some arbitrary matrix $\mathbf{M} \in \mathbb{R}^{n \times d}$ (e.g., matrix of node representations), without explicitly constructing the dense matrix $\exp(\mathbf{A})$.

**Spectral bounds.** Let $\lambda_{\min}(\mathbf{A}) \in \mathbb{R}$ and $\lambda_{\max}(\mathbf{A}) \in \mathbb{R}$ denote the smallest and largest eigenvalues of $\mathbf{A}$. Chebyshev approximation requires an interval $[\lambda_{\min}, \lambda_{\max}]$ containing the spectrum of $\mathbf{A}$. Some potential approaches for obtaining the interval $[\lambda_{\min}, \lambda_{\max}]$ are the following:

- **Degree (Gershgorin) bounds.** For undirected binary graphs, we have that

$$\lambda_{\max}(\mathbf{A}) \leq d_{\max}, \qquad \lambda_{\min}(\mathbf{A}) \geq -d_{\max},$$

  where $d_{\max} = \max_i \sum_j \mathbf{A}_{ij}$ is the maximum (weighted) degree of the graph. The complexity of computing these bounds is $\mathcal{O}(m)$. The main weakness of these bounds is that they might be loose.

- **Power iteration bounds.** Since $\mathbf{A}$ is symmetric, the largest eigenvalue can be estimated by power iteration:

$$\mathbf{v}^{(t+1)} = \frac{\mathbf{A}\mathbf{v}^{(t)}}{\|\mathbf{A}\mathbf{v}^{(t)}\|}$$
$$\hat{\lambda}_{\max} = \mathbf{v}^{(t)\top}\mathbf{A}\mathbf{v}^{(t)}$$

  A symmetric interval can then be chosen as follows:

$$\lambda_{\min} = -\hat{\lambda}_{\max} \qquad \text{and} \qquad \lambda_{\max} = \hat{\lambda}_{\max}.$$

  Power iteration have a complexity of $\mathcal{O}(mT)$ for $T$ iterations.

- **Lanczos bounds.** Alternatively, a Lanczos tridiagonalization of dimension $m$ yields a small tridiagonal matrix $\mathbf{T}_m$ whose extremal eigenvalues approximate those of $\mathbf{A}$. Let $\mu_{\min}$ and $\mu_{\max}$ be the smallest and largest eigenvalues of $\mathbf{T}_m$. We set

$$\lambda_{\min} \approx \mu_{\min} \qquad \text{and} \qquad \lambda_{\max} \approx \mu_{\max}$$

  This provides the tightest bounds at slightly higher preprocessing cost. Specifically, the complexity of Lanczos bounds is $\mathcal{O}(mm_\ell)$ for Krylov dimension $m_\ell$.

**Affine rescaling.** Given the interval $[\lambda_{\min}, \lambda_{\max}]$, we define $\alpha$ and $\beta$ as follows:

$$\alpha = \frac{\lambda_{\max} + \lambda_{\min}}{2},$$
$$\beta = \frac{\lambda_{\max} - \lambda_{\min}}{2}.$$

Then, we have

$$\mathbf{A} = \alpha\mathbf{I} + \beta\mathbf{B},$$
$$\mathbf{B} = \frac{\mathbf{A} - \alpha\mathbf{I}}{\beta},$$

and the spectrum of $\mathbf{B}$ lies in $[-1, 1]$. Consequently,

$$\exp(\mathbf{A}) = \exp(\alpha)\exp(\beta\mathbf{B}).$$

**Chebyshev expansion.**   Let $T_k$ denote Chebyshev polynomials of the first kind:

$$T_0(\mathbf{B}) = \mathbf{I},$$
$$T_1(\mathbf{B}) = \mathbf{B},$$
$$T_{k+1}(\mathbf{B}) = 2\mathbf{B}T_k(\mathbf{B}) - T_{k-1}(\mathbf{B}).$$

The scalar function $f(x) = \exp(\beta x)$ on $[-1, 1]$ admits the truncated expansion

$$f(x) \approx \frac{c_0}{2} + \sum_{k=1}^{K} c_k T_k(x),$$

where the coefficients are given by the cosine–quadrature formula

$$c_k = \frac{2}{N}\sum_{j=0}^{N-1} \exp\big(\beta\cos\theta_j\big)\cos(k\theta_j) \quad \text{with} \quad \theta_j = \frac{\pi(j + \frac{1}{2})}{N}.$$

**Operator form (sparse implementation).**   Instead of forming $\mathbf{B}$ or $T_k(\mathbf{B})$ explicitly, the method applies them to features using sparse matrix–vector products. For any $\mathbf{M} \in \mathbb{R}^{n \times d}$, we have

$$\mathbf{B}\mathbf{M} = \frac{\mathbf{A}\mathbf{M} - \alpha\mathbf{M}}{\beta}.$$

The Chebyshev recursion is evaluated as follows:

$$T_0(\mathbf{B})\mathbf{M} = \mathbf{M},$$
$$T_1(\mathbf{B})\mathbf{M} = \mathbf{B}\mathbf{M},$$
$$T_{k+1}(\mathbf{B})\mathbf{M} = 2\mathbf{B}T_k(\mathbf{B})\mathbf{M} - T_{k-1}(\mathbf{B})\mathbf{M},$$

requiring only repeated sparse multiplications by $\mathbf{A}$.

The final approximation is given by:

$$\exp(\mathbf{A})\mathbf{M} \approx \exp(\alpha)\left(\frac{c_0}{2}\mathbf{M} + \sum_{k=1}^{K} c_k\, T_k(\mathbf{B})\mathbf{M}\right)$$

The time complexity of the Chebyshev approximation is $\mathcal{O}(K\,m\,d)$, while its memory complexity is $\mathcal{O}(m + n\,d)$. Importantly, at no point is a dense $n \times n$ matrix formed, making the method scalable to large graphs. The product $\exp(-\mathbf{A})\mathbf{M}$ for the inverse mapping can be approximated analogously.

## C. Additional Experiments

### C.1. Molecular Perturbation Experiment

Following the experiments conducted by Errica & Niepert (2024) (Figures 3 and 5), we also investigated whether the proposed model can answer probabilistic queries about graphs. Specifically, we trained the INVGNN model on the ogbg-molpcba dataset (Hu et al., 2020) using maximum likelihood estimation. After training, the log-likelihood of each node in

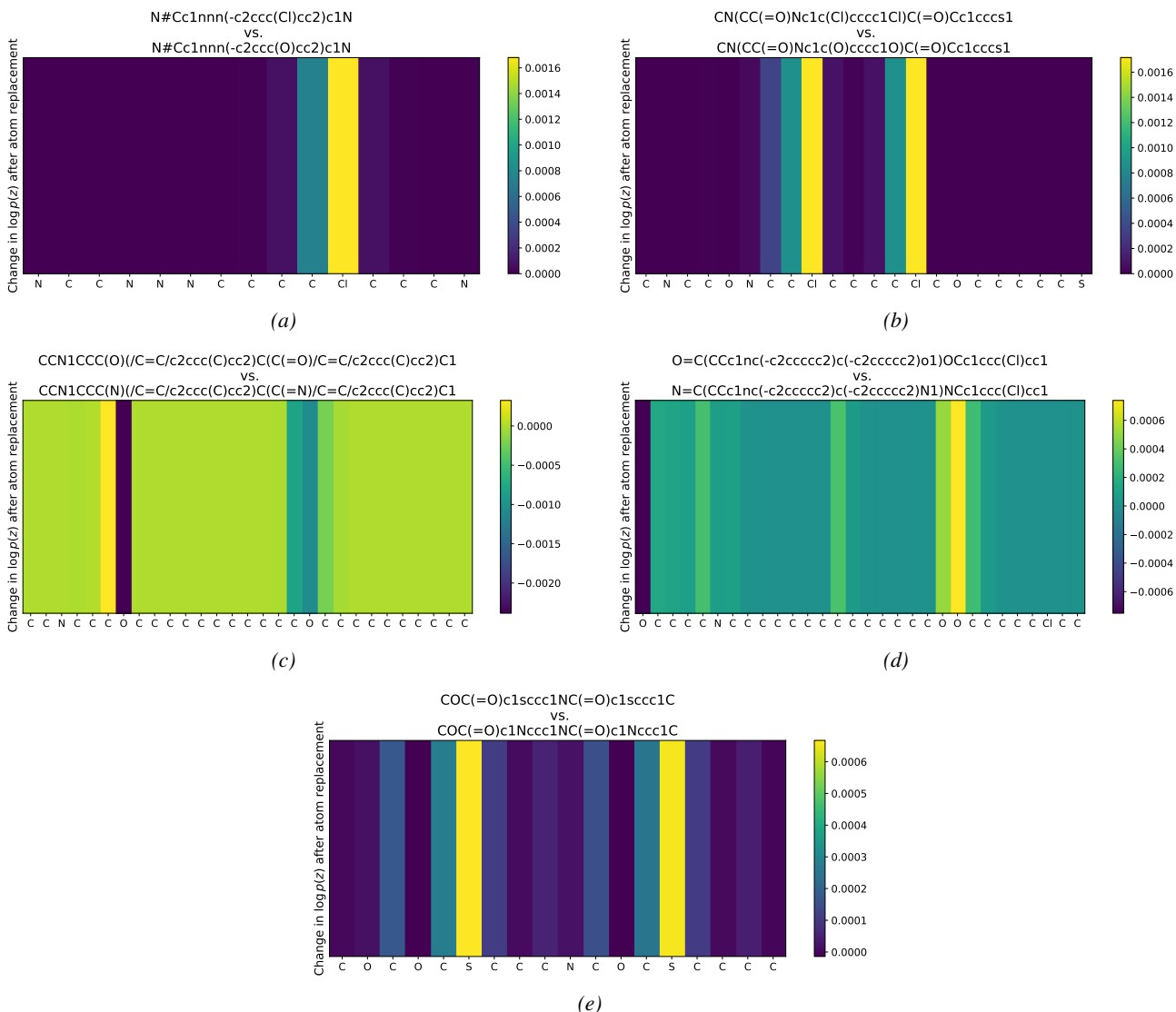

*Figure 5.* The five subplots represent five molecules from ogbg-molpcba. The heatmaps illustrate variations in the log-likelihood of the nodes under specific atomic modifications. Nearby atoms are also affected by these changes.

a graph can be computed via the change-of-variable formula (Equation (1)). We considered the five molecular examples from (Errica & Niepert, 2024), in which certain atoms are replaced in the SMILES representation, and computed the negative log-likelihood for each node before and after replacement. The results are illustrated in Figure 5. Our findings are consistent with those reported by Errica & Niepert (2024). For example, in Figure 5a, where the Chlorine (*Cl*) atom is replaced with Oxygen (*O*), the log-likelihood increases because *Cl* is deactivating and is therefore less likely to occur attached to an all-carbon atom group.

## C.2. Approximation Scheme

We next evaluate the approximation scheme described in Appendix B. This is particularly useful for large graphs where computing the matrix exponential of the adjacency matrix exactly is computationally infeasible. Note that the proposed scheme does not invert the approximation, but it instead approximates the inverse function directly. To quantify the approximation error, we performed an experiment on the MUTAG dataset (Morris et al., 2020). For each graph of MUTAG, we generated an $n \times 32$ random matrix $\mathbf{X}$ (where $n$ denotes the number of nodes of the graph) and approximated the product

*Table 6.* Error of the approximation scheme described in Appendix B as a function of the highest polynomial degree $K$.

| | $K$ | | | | |
|---|---|---|---|---|---|
| | 5 | 10 | 15 | 20 | 25 |
| avg. $\|\mathbf{X} - \hat{\mathbf{X}}\|_F$ ($L=1$) | 1156.0 | 22.6 | 0.253 | $3.78 \times 10^{-4}$ | $5.11 \times 10^{-8}$ |
| avg. $\|\mathbf{H} - \hat{\mathbf{H}}\|_F$ ($L=1$) | 61.0 | 0.932 | 0.012 | $1.25 \times 10^{-5}$ | $1.03 \times 10^{-9}$ |
| avg. $\|\mathbf{X} - \hat{\mathbf{X}}\|_F$ ($L=2$) | 43308.8 | 458.5 | 0.442 | $6.41 \times 10^{-4}$ | $1.58 \times 10^{-7}$ |
| avg. $\|\mathbf{H} - \hat{\mathbf{H}}\|_F$ ($L=2$) | 291.4 | 3.850 | 0.006 | $5.18 \times 10^{-6}$ | $1.67 \times 10^{-9}$ |
| avg. $\|\mathbf{X} - \hat{\mathbf{X}}\|_F$ ($L=3$) | 458631.0 | 1423.0 | 0.516 | $5.97 \times 10^{-4}$ | $3.63 \times 10^{-7}$ |
| avg. $\|\mathbf{H} - \hat{\mathbf{H}}\|_F$ ($L=3$) | 2093.3 | 32.16 | 0.100 | $8.79 \times 10^{-5}$ | $3.10 \times 10^{-8}$ |

$\underbrace{\exp(\mathbf{A}) \ldots \exp(\mathbf{A})}_{L \text{ times}} \mathbf{X}$ using the approach described in Appendix B. Let $\hat{\mathbf{H}}$ denote the approximation output and $\mathbf{H}$ the exact matrix. We then approximated the product $\underbrace{\exp(-\mathbf{A}) \cdots \exp(-\mathbf{A})}_{L \text{ times}} \hat{\mathbf{H}}$, which corresponds to the inverse function, and let $\hat{\mathbf{X}}$ denote the output. Finally, we computed $\|\mathbf{H} - \hat{\mathbf{H}}\|_F$ and $\|\mathbf{X} - \hat{\mathbf{X}}\|_F$, and report their average values for different numbers of Chebyshev terms in Table 6. Power iteration was used to estimate the spectral bounds. It can be observed that for a small number of terms, the approximation error is relatively large. However, the error decreases as $K$ increases. For $K \geq 15$, both the approximate output $\hat{\mathbf{H}}$ and the reconstructed input $\hat{\mathbf{X}}$ closely match their true counterparts. These results indicate that the approximation scheme is reliable and can be confidently applied to datasets where exact computation of the matrix exponential is not feasible.

### C.3. Outlier Detection

We conducted experiments on three real-world datasets, MUTAG, ENZYMES and ZINC. MUTAG and ENZYMES are standard graph classification datasets contained in the TUDataset collection (Morris et al., 2020). ZINC is a popular molecular dataset (Irwin et al., 2012) where the task is to predict the constrained solubility of molecules, an important chemical property for designing generative GNNs for molecules. For our experiments, we used the subset of ZINC that contains 10,000 training samples. For MUTAG and ENZYMES, we treated the samples of one class as "normal" and those of another class as "outliers", while for ZINC, we treated the samples whose target is less than 3 as "outliers" and the remaining samples as "normal". From the normal class, we randomly sampled 20 graphs and removed them from the training set, and we also sampled 20 graphs from the outlier class. We trained the model on the remaining normal graphs using maximum likelihood estimation. Once training was complete, we evaluated the model on the 40 selected graphs (20 normal and 20 outliers) by computing the negative log-likelihood (NLL) of each node. For each graph, we averaged the NLLs of its nodes, and Figure 6 illustrates histograms of these per-graph averages.

In most cases, INVGNN assigns lower NLLs to normal samples than to outliers, indicating that the model captures aspects of the underlying data distribution. However, the model does not perfectly detect outliers. This is not surprising since the classification task on ENZYMES (and on MUTAG for equally sized classes) is known to be challenging, and even models trained on this specific task might achieve suboptimal performance. Furthermore, the training set that contains normal graphs is relatively small, so patterns present in normal graphs from the test set may not be observed during training.

We also consider a different toy scenario. We construct an Erdös-Rényi graph with 20 nodes and edge probability 0.2, and annotate all nodes with feature vectors sampled from $\mathcal{N}(\mathbf{0}, \mathbf{I})$. An INVGNN model consisting of 10 layers is trained on this graph using maximum likelihood estimation. After training, we add a new node to the graph and connect it to 10 randomly chosen existing nodes. This new node is structurally anomalous, since its degree is unusually high compared to the original nodes. The exponential of the new adjacency matrix is computed, and the new node is annotated with a vector also sampled from $\mathcal{N}(\mathbf{0}, \mathbf{I})$. Finally, the graph is fed to the model to obtain negative log-likelihoods for all 21 nodes.

The graph along with the negative log-likelihoods are visualized in Figure 7. The more intense the color of a node, the higher the negative log-likelihood. Note that the newly added node has ID 21, and that the negative log-likelihood of this node is the highest among all nodes. This indicates that the model successfully identifies this node as an outlier.

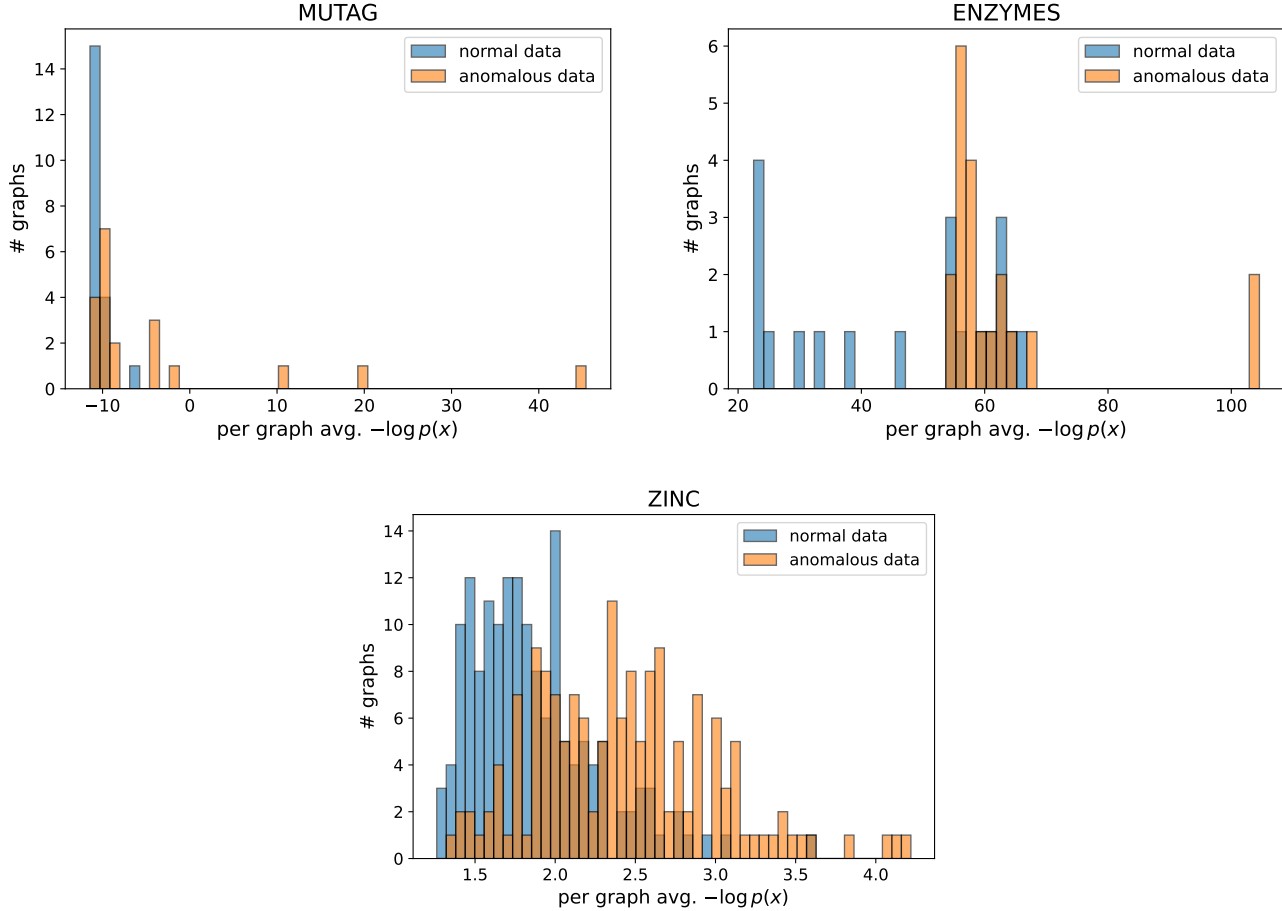

*Figure 6.* Histograms of the average negative log-likelihoods of nodes in 20 normal and 20 anomalous graphs from the MUTAG, ENZYMES, and ZINC datasets.

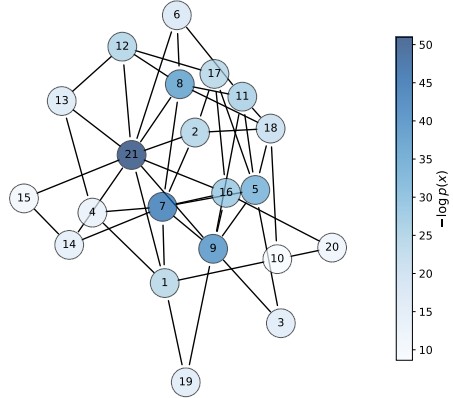

*Figure 7.* Negative log-likelihoods of the nodes of a graph produced by an already-trained INVGNN model. Node 21 was added to the graph post-training and its degree is unusually high compared to the original nodes.

