# OpenReview forum: "InvGNN: Learning Invertible Node Representations on Graphs"
_ICML.cc/2026/Conference — ICML 2026 regular_

### Official Review · Reviewer_DSkJ · 2026-02-19

**Soundness:** 3
**Presentation:** 4
**Significance:** 4
**Originality:** 4
**Overall Recommendation:** 5
**Confidence:** 5

**Summary:**

The paper introduces a strategy to obtain invertible GNNs by means of an exponential operator over the adjacency matrix combined with the use of rotation (invertible) weight matrices and invertible activation functions. The approach allows to obtain a bijective mapping from node features to node embeddings.

**Compliance With Llm Reviewing Policy:**

Affirmed.

**Final Justification:**

All concerns addressed.

**Key Questions For Authors:**

- I was wondering if there are potential works in the literature that already discuss how to make exp(A) sparse while maintaining invertibility, assuming it makes sense from a conceptual point of view.
- Please discuss the advantage/difference of your approach with respect to Liu, Jenny, et al. "Graph normalizing flows." Advances in Neural Information Processing Systems 32 (2019).

**Limitations:**

Yes

**Strengths And Weaknesses:**

Personally, this paper was overdue. Invertible mappings of node embeddings have long eluded the graph ML community, and the authors have done a magnificent job under almost every aspect:
- The introduction is clear, direct, assigns proper credit, and provides a valid motivation for studying this problem
- The paper is honest in surfacing the computational limitations, which is absolutely fine given the significance of the contribution for the graph ML community.
- The paper is mostly self-contained, well-organized, and technically sound. I did not check the proofs in detail, but the mathematical formalization and preliminaries are well presented and one can tell the authors put significant effort on it.
- The paper highlights the different advantages of having an invertible mapping, rather than simply presenting performance metrics. This says a lot about the approach: invertibility is indeed useful for a number of reasons which go beyond solving a task.
- The experiments are clearly described and follow fair and reproducible setups

Reading this paper felt like going back of almost a decade, where the graph ML community focused on simple but impactful ideas. Applying the matrix exponential of adjacency matrix is an extremely original idea and may have a great impact in the years to come.

When considering the quadratic/cubic cost of storing/computing $exp(A)$, one must remember that
- it can be pre-computed for each graph
- and the cost can be approximated

I am confident there will be future room for sparse and invertible approximations of $exp(A)$, which makes the approach even more intriguing in the context of generative models of graphs, for instance.

There are no particular concerns about the paper, rather opportunities for clarifications and improvements:
- line 67-68; typo build -> built
- line 130: please clarify what you mean by 1x1 convolution in this context, it was not immediately clear
- while sections 4.1, 4.2 and 4.4 support the authors’ claims about the effectiveness and necessity of invertible mappings, section 4.3 is possibly the weakest section in terms of presentation and execution. First, readers might not be familiar with the change of variable formulas and symbols like $p(z),f(x)$ have not been properly defined in the section. The authors are encouraged to improve accessibility of this section. Second, in order for me to strongly support this work, I would require a stronger empirical evaluation of outlier detection: the generative process used by the authors, though simple, really feels too artificial and easy to achieve for different kind of networks, for instance the probabilistic GSPN of [1]. The authors could perhaps add a couple of real-world benchmarks to show that artificially modifying graphs (e.g., molecules) leads to worse likelihood scores, such as in [1, Fig. 3].

[1] Errica, Federico, and Mathias Niepert. "Tractable Probabilistic Graph Representation Learning with Graph-Induced Sum-Product Networks." ICLR 2024.

---

> ### Author Rebuttal · Authors · 2026-03-30
>
> We thank the reviewer for the positive evaluation and the constructive feedback. Please find our responses below.
>
> **W1**: We thank the reviewer for pointing out this typo.
>
> **W2**: The term "1x1 convolution" refers to a node-wise linear transformation applied across the feature dimension. This is equivalent to a weight matrix applied independently to each node representation, which allows the model to learn complex feature correlations. The term originates from the literature on normalizing flows, where such layers are typically defined for image data and applied at each spatial location. We will clarify this in the next revision of the manuscript.
>
> **W3a**: This is a fair point. This equation is indeed the well-known change-of-variables formula which forms the basis of normalizing flows. Due to space constraints, we did not provide a detailed explanation of this equation in the original submission. We will include a dedicated section in the Appendix in the revised version.
>
> **W3b**: Following the reviewer's suggestion, we conducted experiments on three real-world datasets, MUTAG, ENZYMES and ZINC. For MUTAG and ENZYMES, we treated the samples of one class as "normal" and those of another class as "outliers", while for ZINC, we treated the samples whose target is less than 3 as "outliers". From the normal class, we randomly sampled 20 graphs and removed them from the training set, and we also sampled 20 graphs from the outlier class. We trained the model on the remaining normal graphs. Once training was complete, we evaluated the model on the 40 selected graphs (20 normal and 20 outliers) by computing the negative log-likelihood (NLL) of each node. For each graph, we averaged the NLLs of its nodes, and histograms of these per-graph averages are illustrated [here](https://anonymous.4open.science/r/InvGNN-D166/outlier_detection/). We also report below the average of the per-graph average node NLLs for each class.
>
> ||MUTAG|ENZYMES|ZINC|
> |-|-|-|-|
> |normal|$-10.4(\pm0.9)$|$46.1(\pm16.2)$|$1.8(\pm0.3)$|
> |outlier|$-3.2(\pm13.5)$|$63.0(\pm14.0)$|$2.3(\pm0.5)$|
>
> In most cases, InvGNN assigns lower NLLs to normal samples than to outliers, indicating that the model captures aspects of the underlying data distribution. However, the model does not perfectly detect outliers. This is not surprising since the classification task on ENZYMES (and on MUTAG for equally sized classes) is known to be challenging, and even models trained on this task achieve suboptimal performance. Furthermore, the training set that contains normal graphs is relatively small, so patterns present in normal graphs from the test set may not be observed during training. We plan to experiment with larger datasets and will update Subsection 4.3 in the next version of the paper to include these results.
>
> We also conducted the experiment from [1, figures 3 and 5] (Errica and Niepert, ICLR'24), as suggested by the reviewer. Specifically, we trained the InvGNN model on the ogbg-molpcba dataset using maximum likelihood estimation. After training, the log-likelihood of each node in a graph can be computed via the change-of-variable formula. We considered the five molecular examples from [1], in which certain atoms are replaced in the SMILES representation, and computed the NLL for each node before and after replacement. The results are available [here](https://anonymous.4open.science/r/InvGNN-D166/ogbg-molpcba/). Our observations are consistent with those reported in [1]. For example, in the first figure (mol1.pdf), where the Chlorine (Cl) atom is replaced with Oxygen (O), the log-likelihood increases because Cl is deactivating and is therefore less likely to occur attached to an all-carbon atom group.
>
> **Q1**: While the Chebyshev method provides efficiency, achieving intrinsic sparsity while guaranteeing invertibility remains an open challenge. One could introduce sparsity by considering only a subgraph of specific radius $k$ around each node, but this would break invertibility. To the best of our knowledge, there is currently no approach that can make $\exp(\mathbf{A})$ sparse while preserving invertibility. We believe that this is an interesting direction and will add a discussion of it as potential future work in the conclusion of the revised manuscript.
>
> **Q2**: In graph normalizing flows, the GNN is used as a non-invertible component within an additive coupling layer. The model is inspired by reversible residual networks and treats the GNN as a "black box", so the overall invertibility of the model relies on the additive coupling structure rather than the GNN itself. In contrast, InvGNN is based on a **by construction invertible** graph operator. That is, the message passing operation is itself bijective, and expressive enough that the model can attain 1-WL power. This design provides a direct, invertible link between the graph topology and the latent space, without relying on the feature-splitting scheme required by coupling layers.

---

> > ### Author Rebuttal · Reviewer_DSkJ · 2026-04-01
> >
> > I thank the authors for their answers. I am satisfied and will champion this paper, I think it is a valuable contribution to the graph ML community.

---

> > > ### Author Response · Authors · 2026-04-02
> > >
> > > We sincerely thank the reviewer for their supportive remarks and constructive suggestions which will help improve our paper

---

### Official Review · Reviewer_7hQH · 2026-03-08

**Soundness:** 3
**Presentation:** 3
**Significance:** 3
**Originality:** 2
**Overall Recommendation:** 5
**Confidence:** 2

**Summary:**

This paper propose InvGNN. InvGNN uses the exponential of adjacency matrix as the graph operator for feature aggregation, and use a invertible 1x1 concolution layer for feature update. The combination of the two make the system invertible. To overcome the computation complexity of exponential computation, the author uses an approximation. The author shows the 1-WL expressivity of the system, and show the potential of using the system for graph generation.

**Compliance With Llm Reviewing Policy:**

Affirmed.

**Final Justification:**

I was positive about the paper's overall contribution, and the rebuttal addresses my concern, and I think this is a solid paper.

**Key Questions For Authors:**

NA

**Limitations:**

Yes

**Strengths And Weaknesses:**

## Strength

Very interesting idea. The invertibility of GNN is largely underexplored, and shifting focus to this can open up new research path, especially for graph generation.

The paper is well-written. The concepts and ideas are easy to follow. The prediction and generation are unified by the same system, and is well-backed by experiments.

## Weaknesses

- The framework is novel, but the components are not new. Using matrix exponential is very common for spectral analysis, and convolution is also well-explored.

- While the author acknowledged the complexity issue, I believe this should still be emphasized. The exact form of the method does not scale. While the approximation alternative proposed by the author is more efficient, it is also not exactly invertible, and hence more discussion on the impact of the approximation can be provided.

- The formulation also requires the graph to be undirected, this is a critical limitation not mentioned by the authors.

- The experiments are in toy setting. The prediction uses outdated and small dataset. Other experiments are in synthetic setting, which puts doubts on the actual utility of the system.

- Frankly, the prediction improvement is merely on par, and usually within standard deviation, but I think is not a big weakness for the purpose of an invertible network.

---

> ### Author Rebuttal · Authors · 2026-03-30
>
> We sincerely thank the reviewer for the positive evaluation and the constructive comments. Below, we address the major comments raised by the reviewer.
>
> **W1**: While the matrix exponential and spectral convolutions are indeed well-established components, their integration into an invertible GNN architecture that preserves 1-WL expressivity represents a novel contribution. Previous works typically employ GNNs as "black-box", non-invertible components within a flow. In contrast, InvGNN is the first to ensure that the **message passing operation itself is bijective**, without compromising the ability to **distinguish non-isomorphic graphs**.
>
> **W2**: While we agree with the reviewer that exact computation of the matrix exponential does not scale to large graphs, we should stress that in several key application domains of graph machine learning, such as chemoinformatics, graphs are relatively small. In these domains, we can compute $\exp(\mathbf{A})$ exactly.
>
> For large graphs, one may apply the approximation scheme described in the Appendix. Note that we do not invert the approximation, but we instead **approximate the inverse function directly**. To quantify the approximation error, we performed an experiment on the MUTAG dataset. For each graph of MUTAG, we generated an $n \times 8$ random matrix $\mathbf{X}$ (where $n$ is the number of nodes) and approximated the product $\exp(\mathbf{A}) \exp(\mathbf{A}) \mathbf{X}$ using the approach described in the Appendix. Let $\hat{\mathbf{H}}$ denote the approximation output and $\mathbf{H}$ the exact matrix. We then approximated the product $\exp(-\mathbf{A}) \exp(-\mathbf{A}) \mathbf{H}$, which corresponds to the inverse function, and let $\hat{\mathbf{X}}$ denote the output. Finally, we computed $|| \mathbf{H} - \hat{\mathbf{H}} ||_F$ and $|| \mathbf{X} - \hat{\mathbf{X}} ||_F$, and report their average values for different numbers of Chebyshev terms in the following Table. Power iteration was used to estimate the spectral bounds.
>
> | | 5 | 10 | 15 | 20 | 25 |
> |-|-|-|-|-|-|
> | avg. $\| \mathbf{X} - \hat{\mathbf{X}} \|_F$ | $37053.9$ | $153.8$ | $0.250$ | $2.83 \times 10^{-4}$ | $7.73 \times 10^{-8}$ |
> | avg. $\| \mathbf{H} - \hat{\mathbf{H}} \|_F$ | $195.3$   | $0.841$ | $3.73 \times 10^{-3}$ | $3.60 \times 10^{-6}$ | $9.73 \times 10^{-10}$ |
>
> It can be observed that for a small number of terms, the approximation error is relatively large. Ηowever, the error decreases as $K$ increases. For $K \geq 15$, both the approximate output $\hat{\mathbf{H}}$ and the reconstructed input $\hat{\mathbf{X}}$ closely match their true counterparts. These results indicate that the approximation scheme is reliable and can be confidently applied to datasets where exact computation of the matrix exponential is not feasible.
>
> **W3**: We would like to clarify that our formulation **does not require** the graph to be undirected. The matrix exponential $\exp(\mathbf{A})$ and its inverse are also defined for directed graphs. In this setting, the element of the $i$-th row and $j$-th column of $\exp(\mathbf{A})$ is equal to the weighted sum of all **directed** walks of all lengths between nodes $v_i$ and $v_j$. However, since $\mathbf{A}$ is not symmetric, its eigenvectors may not be well-conditioned and this can lead to numerical instabilities. Instead of computing $\exp(\mathbf{A})$ using the eigenvalue decomposition, we can approximate some product $\exp(\mathbf{A}) \mathbf{x}$ using Krylov subspace methods. We will explicitly clarify this and provide more details in the revised version of the manuscript.
>
> **W4**: The main goal of the experiments was to **demonstrate versatility** across a diverse set of tasks (classification, outlier detection, and generation) rather than pursuing state-of-the-art performance on a single dataset or task. Furthermore, in the graph classification task, we aimed to show that InvGNN matches the performance of standard expressive models like GIN, and that invertibility does not compromise its performance. With regard to dataset scale, we note that we included ogbg-molhiv, a large-scale benchmark, to demonstrate that InvGNN scales beyond toy settings. Finally, in the outlier detection setting, we conducted additional experiments on real-world datasets. Please refer to our response to point W3b of Reviewer DSkJ for the results.

---

> > ### Author Rebuttal · Reviewer_7hQH · 2026-04-03
> >
> > Thanks for addressing my concerns. I still believe this is a solid paper, and I am happy to raise my score (4->5).

---

> > > ### Author Response · Authors · 2026-04-03
> > >
> > > We thank the reviewer for reading our rebuttal and for their thoughtful comments, which we greatly appreciate.

---

### Official Review · Reviewer_iUV8 · 2026-03-11

**Soundness:** 3
**Presentation:** 3
**Significance:** 3
**Originality:** 4
**Overall Recommendation:** 4
**Confidence:** 4

**Summary:**

**The main goal of this work is to design an invertible graph neural network (GNN) that can match the 1-WL expressivity of MPNNs, and show interesting use-cases.** An invertible GNN is exactly what it sounds like: a function on graphs represented by GNNs that can be fully inverted (in the bijective function sense). This has interesting applications in density estimation for outlier detection, generative modeling for generating graph features, and explaining why the prediction is failing on some test samples. The invertible GNN proposed in this work, termed InvGNN, is designed by composing novel invertible GNN layers with invertible activation functions. The invertible GNN layer is designed by combining (a) an invertible aggregation function for aggregating neighborhood information, and (b) invertible neural weights taken from previous work by Kingma & Dhariwal, 2018. The invertible aggregation scheme is constructed by constructing an invertible graph operator, specifically the matrix exponential of the symmetric adjacency matrix for undirected graphs. The invertible activation is Tanh, LeakyReLU, Sigmoid, etc.

The authors theoretically show that a single layer as defined above is not as expressive as a 1-WL test, but stacked layers close this gap in the same way that stacked layers of a GIN closes the gap to a full iterative 1-WL graph isomorphism test.

Authors benchmark InvGNN 7 datasets, against 14 baseline models (although 7 of these were used on 6 datasets belonging to TUDataset collection and the other 7 were used on OGB-molhiv). InvGNN performs decently, it is competitive in 5 datasets, while it falls behind the best method by a substantial margin on ENZYMES and NCI1 datasets.

Experiments on toy datasets for expressive power show that InvGNN can distinguish structurally different nodes. Authors also demonstrate how InvGNN can be used for (a) outlier detection, (b) node feature generation, and (c) decision explanation.

InvGNN has a high time and memory complexity as compared to standard MPNNs, and the authors address that. In the appendix, they provide ways to make this process more efficient, both time-wise and memory-wise.

---
This is a novel and impactful work, with potential impact across several important fields that use graph methodologies, such as in drug molecule generation, anomaly detection in financial networks, and potentially advancing explainability in all domains that employ GNNs.

**Compliance With Llm Reviewing Policy:**

Affirmed.

**Final Justification:**

Invertible GNNs represent a novel and promising direction, enabling applications such as uncertainty estimation and graph probability density estimation. Considering these compelling applications, together with the sound theoretical grounding of the work, I have revised my score in favor of acceptance.

**Key Questions For Authors:**

See above

**Limitations:**

Yes

**Strengths And Weaknesses:**

## Plus and minus points

### In terms of experimental design:
+1 for reporting standard deviation across multiple runs
**-1 for taking results from paper and not ensuring that experimental setup is fair for a fair comparison (unsound experiment design)**
+1 for sharing code
+1 for kinds of experiments performed (outlier, feature gen, etc.)
+1 section 4.5 experiment is very nice

### In terms of presentation:
**-1 for missing equation numbers**

### In terms of addressing limitations:
+1 for approximation methods provided, but it would be good to have results of how good these approximations are.



## Questions

-   Invertible layers do not discard information. Is this sub-optimal in some situations?
-   Which one is the equation number 3 mentioned in line number 175-176?
-   Another two typos in Ln 177-179. d' is not introduced. It should be
    \(v'\in V'\) and not \(v\in V\).
-   Why are different baseline methods used for ogbg-molhiv?
-   Why does LeakyReLU InvGNN despite being invertible still fail on
    some cases to distinguish structurally different nodes that 1-WL can
    detect?
-   Where does the equation in Ln 304 come from?
-   How to incorporate Dropout, BatchNorm, etc.?
-   What happens for directed graphs?



## Suggestions

-   Include equation numbers.
-   For table 1 and table 2, add a row of delta (difference) with the
    best in case InvGNN isn't the best, or with the second
    best. Highlight in colors (green for increase, red for
    decrease). And also highlight the best/second best so that one
    can easily find which two numbers are used to compute the delta.
-   In section 4.3, what should be obvious to an expert audience, but
    can still benefit from added clarity is that higher negative log
    likelihood means higher chance of being an outlier.
-   The exponential map is a dense operator, usually computed via
    eigenvalue decomposition. This is therefore expensive to compute,
    and may not scale to large graphs. Can this be approximated, for
    example via random walks? This should also be part of the future
    work. ****Appendix B discusses some solutions, but this should still be left open for future for discovering more scalable methods (random walks? Markov Chain Monte Carlo estimation methods?)****.



## Discussion

-   The matrix exponential is suggested as an invertible operator. This
    immediately connects me to Lie Algebra and how the exponential map
    takes one from the Lie Algebra to the corresponding continuous
    group. This connection may be important for future researchers, and
    can be part of the future work. For example, ask researchers to
    study this from the perspective of Lie Theory to derive useful
    results.



## Possible issues that I foresee

-   Since the results for baselines have been taken from what is
    reported in their papers, their experimental setup and yours might
    differ (for example the 10-fold cross-validation for TUDatasets used
    by you, or batch sizes). This may mean that the comparison is unfair
    and not be representative of what might happen in real scenarios.
-   The toy examples used in section 4.3, 4.4 are very small. How well
    can this extend to larger datasets?
-   For full invertability, the final layer has to be of the same
    dimension as the feature dimension. What happens when number of
    classes is less (as in most real cases)? What about when number of
    features is less? In general a multiplication of a vector with a
    tall matrix can never be surjective and a wide matrix can never be
    injective.

# Reason for my score
The main reason for this score is that I am not sure which equation does theorem 3.2 refer to, and I am not sure that the experimental comparison in tables 1 and 2 are fair (since results were taken from the papers themselves, which may have different experimental setup).

---

> ### Author Rebuttal · Authors · 2026-03-30
>
> We thank the reviewer for their thoughtful evaluation and for the suggestions, which we will incorporate in the next version of the paper. Below, we provide detailed responses to the major points raised. We use Ik, Qk, and Sk to denote the k-th Issue, Question and Suggestion raised by the reviewer, respectively.
>
> **I1**: We believe there is a misunderstanding here. We would like to clarify that our experimental design **ensures a fair comparison**. Specifically, we adopted the experimental protocol proposed in [1] and used exactly the same train/validation/test splits for each dataset as in [1]. Because the data splits and evaluation metrics are identical, the reported numbers are directly comparable. For this reason, we report the results as presented in that paper and in other works that have followed the same protocol, such as the papers that propose SPN and Nested GNN. In addition, the ogbg-molhiv dataset comes with standard train/validation/test splits. Therefore, all models for which we report performance have been trained and evaluated on exactly the same data splits as InvGNN.
>
> **I2**: We conducted additional experiments on real datasets in response to the reviewers' comments. Please refer to our response to point W3b of Reviewer DSkJ for more details.
>
> **I3**: Let $n$ denote the number of input features and $m$ the number of classes. If $m<n$, we can add $n−m$ dummy classes that never occur in practice. The model will learn to never predict these classes. This is exactly what we did in the experiment described in subsection 4.5. On the other hand, if $n<m$, we can add $m−n$ dummy features with values set to zero.
>
> **Q1**: Indeed, there exist such situations. Consider a task where we would like the model to focus mainly on a subset of features (e.g., object detection in an image where the background is irrelevant). Standard neural networks employ pooling and bottleneck layers which allow them to discard uninformative features. In invertible models, however, such features can never be fully discarded.
>
> **Q2**: We apologize for the omission of the equation numbers. Equation (3) is the one that describes **how node representations are updated** within the InvGNN model and is presented in L135:
> $
> \mathbf{h}_i^{(k)} = \sum_{j=1}^n [\exp(\mathbf{A})]_{ij}  f( \mathbf{h}_j^{(k-1)} \mathbf{W}^{(k)} + \mathbf{b}^{(k)})
> $
> We will ensure that equation numbers are correctly included in the next version of the paper.
>
> **Q3**: We thank the reviewer for bringing these points to our attention. Indeed, we will replace $v \in V'$ with $v' \in V'$. Here, $d'$ simply denotes the hidden dimension size of the model.
>
> **Q4**: Not all models included in our initial experiments on the datasets from the TUDatasets collection have been evaluated on ogbg-molhiv. Therefore, for ogbg-molhiv, we selected a separate set of models that have been reported on this benchmark using its standard splits, ensuring a fair comparison by using the same evaluation protocol.
>
> **Q5**: Invertibility does not imply 1-WL expressive power. Theorem 3.2 states that models that utilize analytic non-polynomial activation functions can achieve this expressive power almost surely. Since LeakyReLU **is not** an analytic non-polynomial activation function, the conclusions of Theorem 3.2 do not apply to this function, and this is verified empirically in subsection 4.2.
>
> **Q6**: This equation is the well-known change-of-variables formula, which forms the basis of normalizing flow models (see section 2 in [2] and section 2 in [3]). It describes how to compute the log-density of a variable $\mathbf{x}$ after applying an invertible transformation. Due to space constraints, we did not provide a detailed explanation of this equation in the original submission. In the revised version, we will include a section in the Appendix that provides background on normalizing flows.
>
> **Q7**: Unfortunately, Dropout cannot be directly employed within invertible architectures since it introduces stochastic, non-invertible transformations. Instead, regularization can be achieved through other techniques such as weight decay. With regard to the normalization layers, it is common practice in invertible models to use ActNorm (instead of BatchNorm), a deterministic per-channel affine transformation initialized from data which preserves invertibility.
>
> **Q8**: Due to space constraints, we refer the reviewer to our response to point W3 of Reviewer 7hQH.
>
> **S4**: We evaluated the proposed approximation scheme and further details can be found in our response to point W2 of Reviewer 7hQH. The development of more scalable methods could indeed be an important direction for future research.
>
> [1] F. Errica et al. "A Fair Comparison of Graph Neural Networks for Graph Classification", In ICLR'20.\
> [2] L. Dinh et al. "Density estimation using Real NVP", In ICLR'17.\
> [3] D.P. Kingma, and P. Dhariwal, "Glow: Generative Flow with Invertible 1x1 Convolutions", In NeurIPS'18.

---

> > ### Author Rebuttal · Reviewer_iUV8 · 2026-03-31
> >
> > All my concerns have been adequately addressed.

---

> > > ### Author Response · Authors · 2026-04-02
> > >
> > > We would like to thank the reviewer once again for their constructive feedback and also for increasing their rating after considering our rebuttal.

---

### Official Review · Reviewer_UM6K · 2026-03-12

**Soundness:** 3
**Presentation:** 2
**Significance:** 3
**Originality:** 3
**Overall Recommendation:** 5
**Confidence:** 2

**Summary:**

This paper proposes a simple invertible aggregation operator based on the exponential of the adjacency matrix (which takes cubic preprocessing time, and is dense so quadratic during training), and an invertible linear transformation from the literature. While it is not 1-WL expressive for a single layer, with more layers it almost always is. It doesn't improve on the GNN standard tasks, but it is also not detrimental, and it has some benefits such as being able to recover explanations in the input feature space, and for generation, though this is not much explored.

**Compliance With Llm Reviewing Policy:**

Affirmed.

**Final Justification:**

No significant concerns remain on my side such that they would reject the paper. As mentioned in the response, the scope is a bit narrow, but the contribution is sound.

**Key Questions For Authors:**

1. How can precision affect the results, given that the exponential sends values to greater ranges?
2. Why could invertibility compromise model expressiveness, if it keeps more information?
3. Relatedly, is there any setting where it is expected to be worse than a standard GNN? Where would we need to not be invertible and why?
4. Usually in classifying tasks the model needs to get rid of data in order to learn efficient or generalizable (non-overfitting) representations, why is it not here?
5. I don't understand why Theorem 3.2 can assume any K, but one can't take K=1 because otherwise it contradicts Prop 3.1. Can you explain this better?
6. Have you tried the efficient approximation of the exponential in the experiments?
7. Have you considered real world experiments for finding outliers?
8. Are there other methods for the tasks "beyond accuracy" that are presented in the paper? It looks like this is the only way that they could be solved, which I doubt.

**Limitations:**

The clarity of theorem 3.2 has to improve, and I don't only mean the statement, but also why it is there in that form and what is the nuance with it. It would also be nice to strengthen the outliers section with a real world task.

**Strengths And Weaknesses:**

Strengths: The non-standard tasks and the possible implications for future work is where this work is interesting. It is a neat construction albeit simple. The implications for interpretability are also quite remarkable.

Weaknesses:
1. Clearly, the fact that the adjacency matrix becomes dense is a big drawback. For instance, many methods that are also quadratic are much better than this one from an accuracy perspective. So this is required to excel in other terms, such as in tasks where invertibility is required. This zone might be narrow, and the paper does not entirely explain why there are not other methods for these other tasks.
2. How does the dense adjacency matrix affect the learned weights? How does the reparameterization(s) of both affect the implicit bias of the model? Are the solutions or activations also less sparse?
3. The theorem 3.2 is really "dense" and not understandable. The assumptions are quite different than the previous proposition, and they are not introduced or explained to see why they are relevant or different or necessary. We don't get a good insight into why the theorem has to take this form or why there needs to be several layers for expressiveness.
4. Given that the new tasks are where the method is best, the outliers section is very synthetic and not realistic.
5. Minor: "sinlge" in l152.

---

> ### Author Rebuttal · Authors · 2026-03-30
>
> We thank the reviewer for the positive assessment and the thoughtful comments. We provide responses to the reviewer's comments and questions below.
>
> **W1 & Q8**: The literature on invertible GNN models is generally limited. The graph operators employed by existing GNNs are not inherently invertible. Invertibility can still be achieved through alternative frameworks (e.g., graph normalizing flows), even when the underlying graph operator is not invertible. Invertible models enable a broader range of applications compared to non-invertible ones. These tasks can potentially be addressed by other methods (e.g., density estimation via kernel density estimation with graph kernels). We emphasize that the objective of our experiments is not to achieve state-of-the-art performance on each individual task or to compare InvGNN against methods specifically tailored to them. Instead, our goal is to **validate the invertibility and expressiveness** of the proposed model, and to demonstrate its **versatility across multiple tasks**. In summary, while alternative methods exist for some tasks, comprehensive comparisons with task-specific approaches are beyond the scope of this work.
>
> **W2 & Q1**: The density of the adjacency matrix is not directly related to the density of the learned representations. For example, a complete graph does not necessarily yield denser representations than other graphs. However, the proposed graph operator can produce representations with higher norms, which may lead to exploding gradients, often resulting in unstable or divergent training. To address this, as discussed in subsection 4.1, we scale the adjacency matrix before computing $\exp(\mathbf{A})$ by dividing each element by the largest eigenvalue or in the case of multiple graphs, by the average of the largest eigenvalues across all graphs.
>
> **W3 & Q5**: We believe there may be some misunderstanding here. Proposition 3.1 shows that a single layer of the proposed model can fail to distinguish non-isomorphic nodes that can be distinguished by **multiple iterations** of 1-WL. Our hypothesis was that since $\exp⁡(\mathbf{A})$ encodes relationships not only between immediate neighbors but also between nodes that are far apart in the graph, a single layer of the model could be sufficient.
>
> Theorem 3.2 states that an **InvGNN with $K$ layers** can, almost surely, distinguish any pair of non-isomorphic nodes that 1-WL with $K$ iterations can distinguish. The main conditions are that the model must use analytic non-polynomial activation functions and that the hidden dimension size must be at least $2n+1$ where $n$ is the number of nodes in the largest graph. These conditions arise because the proof relies on a theorem from Amir et al. (restated as Theorem A.4 in the Appendix), which guarantees that functions on spaces of measures of a specific form are injective. In our setting, we designed the aggregation mechanism of InvGNN to satisfy this form.
>
> **W4 & Q7**: We conducted additional experiments on real datasets. Please see our response to Reviewer DSkJ (point W3b).
>
> **Q2**: **Invertibility does not guarantee expressiveness**. For example, the identity matrix is invertible but using it as a graph operator provides no expressiveness since it completely ignores the graph structure. To be expressive, the operator must also appropriately encode the structure of graphs.
>
> **Q3**: Yes, such settings do exist. For graphs characterized by strong homophily, GCN may exhibit superior performance because of its normalization scheme which smooths node representations. In contrast, the aggregation mechanism of InvGNN which incorporates information from a larger number of nodes might introduce noise. As a result, it can dilute the local similarity signal that is crucial in homophilic graphs.
>
> **Q4**: In our setting, matrix $\exp(\mathbf{A})$ is not part of the model's learnable parameters. It instead serves as a **graph operator to propagate information**. By providing the model with information about the number of walks of arbitrary length between nodes (encoded in $\exp(\mathbf{A})$), the model can more effectively capture structural relationships and node proximities. For instance, if an element of $\exp(\mathbf{A})$ is 0, this indicates that the corresponding nodes belong to different connected components. This information cannot be directly inferred from the adjacency matrix.
>
> **Q6**: We have indeed implemented the model that uses the proposed approximation scheme and verified that it produces comparable classification results on small datasets. However, the current implementation processes one graph at a time instead of batches of graphs and is slow. We plan to develop a batched version and compare it against the exact model. We will report the results in the next version of the paper. Note that we also evaluated the quality of the approximation offered by the proposed scheme and the results are provided in our response to point W2 of Reviewer 7hQH.

---

> > ### Author Rebuttal · Reviewer_UM6K · 2026-04-01
> >
> > Thank you for the responses. I am satisfied with most answers, especially on the theorem (which I hope can get a bit more context in the revision) and the new experiments, which I also hope to see there. I am not entirely satisfied with the scope of the paper as I would like it compared to more task-specific recipes, but I don't think it is a reason for rejection. I will adjust my score accordingly.

---

> > > ### Author Response · Authors · 2026-04-02
> > >
> > > We sincerely thank the reviewer for taking the time to read our rebuttal, for the very helpful feedback, and for raising their score.

---

### Decision · Program_Chairs · 2026-04-30

**Decision:**

Accept (regular)

**Comment:**

This paper introduces a principled invertible GNN architecture. The authors provide both theoretical guarantees (matching 1-WL expressivity) and practical utility.  This is a novel contribution that all four reviewers agree opens a promising research direction. While scalability remains a limitation, the reviewers unanimously support acceptance after a thorough rebuttal that addressed all raised concerns, and I agree with their assessment.